# Toward a Unified Theory of Gradient Descent under Generalized Smoothness

**Alexander Tyurin** [1] [2]

## Abstract

We study the classical optimization problem $\min_{x \in \mathbb{R}^d} f(x)$ and analyze the gradient descent (GD) method in both nonconvex and convex settings. It is well-known that, under the $L$–smoothness assumption ($\|\nabla^2 f(x)\| \leq L$), the optimal point minimizing the quadratic upper bound $f(x_k) + \langle \nabla f(x_k), x_{k+1} - x_k \rangle + L/2 \|x_{k+1} - x_k\|^2$ is $x_{k+1} = x_k - \gamma_k \nabla f(x_k)$ with step size $\gamma_k = 1/L$. Surprisingly, a similar result can be derived under the $\ell$-generalized smoothness assumption ($\|\nabla^2 f(x)\| \leq \ell(\|\nabla f(x)\|)$). In this case, we derive the step size

$$\gamma_k = \int_0^1 \frac{dv}{\ell(\|\nabla f(x_k)\| + \|\nabla f(x_k)\| \, v)}.$$

Using this step size rule, we improve upon existing theoretical convergence rates and obtain new results in several previously unexplored setups.

## 1. Introduction

We consider optimization problems of the form

$$\min_{x \in \mathbb{R}^d} f(x), \tag{1}$$

where $f : \mathbb{R}^d \to \mathbb{R} \cup \{\infty\}$. Our goal is to find an $\varepsilon$–stationary point, $\bar{x} \in \mathbb{R}^d$ such that $\|\nabla f(\bar{x})\|^2 \leq \varepsilon$, in the nonconvex setting, and an $\varepsilon$-solution, $\bar{x} \in \mathbb{R}^d$ such that $f(\bar{x}) - \inf_{x \in \mathbb{R}^d} f(x) \leq \varepsilon$, in the convex setting. We define $\mathcal{X} = \{x \in \mathbb{R}^d \mid f(x) < \infty\}$, and assume that $\mathcal{X}$ is open convex, $f$ is smooth on $\mathcal{X}$, and continuous on the closure of $\mathcal{X}$.

This is a classical and well-studied problem in optimization (Nesterov, 2018; Lan, 2020). We investigate arguably the

---

[1]AIRI, Moscow, Russia [2]Skoltech, Moscow, Russia. Correspondence to: Alexander Tyurin <alexandertiurin@gmail.com>.

*Proceedings of the 42nd International Conference on Machine Learning*, Vancouver, Canada. PMLR 267, 2025. Copyright 2025 by the author(s).

most popular numerical first-order method, gradient descent (GD):

$$x_{k+1} = x_k - \gamma_k \nabla f(x_k), \tag{2}$$

where $x_0$ is starting point, and $\{\gamma_k\}$ are step sizes.

GD is pretty well understood under the traditional $L$–smoothness assumption, i.e., $\|\nabla f(x) - \nabla f(y)\| \leq L \|x - y\|$ or $\|\nabla^2 f(x)\| \leq L$ for all $x, y \in \mathcal{X}$. However, instead of $L$–smoothness, we investigate $\ell$–smoothness, i.e.,

$$\|\nabla^2 f(x)\| \leq \ell(\|\nabla f(x)\|) \tag{3}$$

for all $x \in \mathcal{X}$ (see Assumption 3.1), where $\ell$ is any non-decreasing positive locally Lipchitz function. This is a recent assumption that captures a much wider set of functions (Li et al., 2024a). In particular, if $\ell(s) = L$, then we get $L$–smoothness. If $\ell(s) = L_0 + L_1 s$, then we get $(L_0, L_1)$–smoothness (Zhang et al., 2019). $L$–smoothness can typically capture "quadratic–like" functions and does not include even $f(x) = x^p$ for $p > 2$ or $f(x) = e^x$. A similar problem arises with $(L_0, L_1)$–smoothness: it does not include $f(x) = -\log x$, for instance. The class of $\ell$–smooth functions can include all these examples and more (Li et al., 2024a).

### 1.1. Related work

$L$**–smoothness:** Under $L$–smoothness, it is well-known that GD converges to an $\varepsilon$–stationary point after $\mathcal{O}(L\Delta/\varepsilon)$, and to an $\varepsilon$-solution after $\mathcal{O}(LR^2/\varepsilon)$ iterations (Nesterov, 2018), where $\Delta := f(x_0) - f^*$, $R := \|x_0 - x_*\|$, $f^* = \inf_{x \in \mathbb{R}^d} f(x)$, and $x_* \in \mathbb{R}^d$ is a solution of (1). Interestingly, this result can be further improved by employing non-constant step sizes (Altschuler & Parrilo, 2024; Grimmer et al., 2024), although relying on the $L$–smoothness assumption.

$(L_0, L_1)$**–smoothness and nonconvex setup:** In the nonconvex setting, Zhang et al. (2019) analyzed the $(L_0, L_1)$–smoothness assumption, which is stated for twice differentiable functions as $\|\nabla^2 f(x)\| \leq L_0 + L_1 \|\nabla f(x)\|$ for all $x \in \mathcal{X}$. Zhang et al. (2019) showed that a clipped version of GD, i.e., (2) with appropriately chosen step sizes, finds an $\varepsilon$–stationary point after $\mathcal{O}(L_0\Delta/\varepsilon + L_1^2\Delta/L_0)$ iterations. Subsequently, Crawshaw et al. (2022); Chen

Table 1: Summary of convergence rates for various GD methods under generalized smoothness assumptions in the **nonconvex** setting. Abbreviations: $R := \|x_0 - x_*\|$, $\Delta := f(x_0) - f(x_*)$, $\varepsilon$ = error tolerance. With the green color we highlight our new results and improvements.

| Setting | Rate | References |
|---|---|---|
| $L$–Smoothness $\left(\|\nabla^2 f(x)\| \leq L\right)$ | $\frac{L\Delta}{\varepsilon}$ | — |
| $(L_0, L_1)$–Smoothness $\left(\|\nabla^2 f(x)\| \leq L_0 + L_1\|\nabla f(x)\|\right)$ | $\frac{L_0\Delta}{\varepsilon} + \frac{L_1\Delta}{\sqrt{\varepsilon}}$ | (Vankov et al., 2024) |
| $(\rho, L_0, L_1)$–Smoothness with $0 \leq \rho \leq 2$ $\left(\|\nabla^2 f(x)\| \leq L_0 + L_1\|\nabla f(x)\|^\rho\right)$ | $\frac{L_0\Delta + L_1\|\nabla f(x_0)\|^\rho \Delta}{\varepsilon}$ | (Li et al., 2024a) (no guarantees when $\rho = 2$) |
| | $\frac{L_0\Delta}{\varepsilon} + \frac{L_1\Delta}{\varepsilon^{(2-\rho)/2}}$ | Sec. 5.3 (**new**) ($L_0\Delta/\varepsilon + L_1\Delta$ when $\rho = 2$) |
| $(\rho, L_0, L_1)$–Smoothness with $\rho > 2$ | $\frac{L_0\Delta}{\varepsilon} + L_1\Delta(2M)^{\rho-2}$ | Sec. 6.2 (**new**) (requires A.6.1) |
| Exponential growth of $\ell$ $\left(\|\nabla^2 f(x)\| \leq L_0 + L_1\|\nabla f(x)\|^2 \exp(\|\nabla f(x)\|)\right)$ | $\frac{L_0\Delta}{\varepsilon} + L_1\Delta e^{2M}$ | Sec. 6.1 (**new**) (requires A.6.1) |

et al. (2023); Wang et al. (2023); Koloskova et al. (2023); Li et al. (2024a;b); Hübler et al. (2024); Vankov et al. (2024) considered the same setup, where the state-of-the-art theoretical iteration complexity $\mathcal{O}\left(L_0\Delta/\varepsilon + L_1\Delta/\sqrt{\varepsilon}\right)$ is obtained by Vankov et al. (2024), without requiring a bounded gradient assumption, dependence on $\|\nabla f(x_0)\|$, or $L$-smoothness of $f$.

$(L_0, L_1)$**–smoothness and convex setup:** Convex problems were investigated by Koloskova et al. (2023). They showed that (2) with normalized step sizes guarantees $\mathcal{O}\left(L_0 R^2/\varepsilon + \sqrt{L/\varepsilon}L_1 R^2\right)$ rate, which requires the $L$–smoothness of $f$. Li et al. (2024a), using GD and accelerated GD (Nesterov, 1983), obtained $\mathcal{O}\left((L_0+L_1\|\nabla f(x_0)\|)R^2/\varepsilon\right)$ and $\mathcal{O}\left(\sqrt{(L_0+L_1\|\nabla f(x_0)\|)R^2/\varepsilon}\right)$, which depend on the norm of the initial gradient $\|\nabla f(x_0)\|$. Furthermore, Takezawa et al. (2024) get $\mathcal{O}\left(L_0 R^2/\varepsilon + \sqrt{L/\varepsilon}L_1 R^2\right)$ rate using the Polyak step size (Polyak, 1987), which also requires $L$–smoothness. Using adaptive step sizes, Gorbunov et al. (2024); Vankov et al. (2024) concurrently provided the convergence rate $\mathcal{O}\left(L_0 R^2/\varepsilon + L_1^2 R^2\right)$. Also, with a fixed step size, (Li et al., 2024a) proved the convergence rate $\mathcal{O}\left(L_0 R^2/\varepsilon + L_1\|\nabla f(x_0)\|^2 R^2/\varepsilon\right)$. Gorbunov et al. (2024); Vankov et al. (2024) also analyzed accelerated versions of GD, but Gorbunov et al. (2024) get an exponential dependence on $L_1$ and $R$, while Vankov et al. (2024) requires solving an auxiliary one-dimensional optimization problem in each iteration.

$\ell$**–smoothness:** Li et al. (2024a) introduced the $\ell$–smoothness assumption and analyzed GD as well as an accelerated version of GD. They obtained convergence rates of $\mathcal{O}\left(\ell(\|\nabla f(x_0)\|)R^2/\varepsilon\right)$ and $\mathcal{O}\left(\sqrt{\ell(\|\nabla f(x_0)\|)R^2/\varepsilon}\right)$ for GD and accelerated GD in the convex setting, and a rate of $\mathcal{O}\left(\ell(\|\nabla f(x_0)\|)\Delta/\varepsilon\right)$ in the nonconvex setting for GD. In all cases, their results depend on $\ell(\|\nabla f(x_0)\|)/\varepsilon$ and require that the function $\ell(s)$ grows more slowly than $s^2$ in the nonconvex setting. For instance, they cannot guarantee convergence when $\ell(s) = L_0 + L_1 s^2$. The $\ell$–smoothness assumption was also consider in online learning (Xie et al., 2024).

### 1.2. Contributions

We take the next step in the theoretical understanding of optimization for $\ell$–smooth functions. Using new bounds, we discover a new step size rule in Algorithm 1, which not only improves previous theoretical guarantees but also provides results in settings where GD's convergence was previously unknown (see Tables 1 and 2). In particular,

• We prove new key auxiliary results, Lemmas 4.3 and 4.5, which generalize the classical results and allow us to derive the step size $\gamma_k = \int_0^1 \frac{dv}{\ell(\|\nabla f(x_k)\| + \|\nabla f(x_k)\|v)}$.

• Using this step size, we significantly improve the convergence rate established by Li et al. (2024a) in the nonconvex setting. For instance, under the $(\rho, L_0, L_1)$–smooth assumption with $0 \leq \rho < 2$, we improve their convergence rate of GD from $\frac{L_0\Delta}{\varepsilon} + \frac{L_1\|\nabla f(x_0)\|^\rho \Delta}{\varepsilon}$ to $\frac{L_0\Delta}{\varepsilon} + \frac{L_1\Delta}{\varepsilon^{(2-\rho)/2}}$. Moreover, Li et al. (2024a) do not guarantee the convergence of GD when $\rho = 2$, while our theory, for the first time, establishes the rate $\frac{L_0\Delta}{\varepsilon} + L_1\Delta$, which we believe is important in view of the motivating examples from Section 2. Despite the $(\rho, L_0, L_1)$–smoothness assumption, our theory remains

Table 2: Summary of convergence rates for various GD methods under generalized smoothness assumptions in the **convex** setting. Abbreviations: $R := \|x_0 - x_*\|$, $\Delta := f(x_0) - f(x_*)$, $M_0 := \|\nabla f(x_0)\|$, $\varepsilon =$ error tolerance. With the green color we highlight our new results and improvements.

| Setting | Rate | References |
|---|---|---|
| $L$–Smoothness | $\frac{LR^2}{\varepsilon}$ | ——[a] |
| $(L_0, L_1)$–Smoothness | $\frac{L_0 R^2}{\varepsilon} + \min\{L_1^2 R^2, \frac{L_1 M_0 R^2}{\varepsilon}\}$ | (Li et al., 2024a) (Gorbunov et al., 2024) (Vankov et al., 2024) |
| | $\frac{L_0 R^2}{\varepsilon} + \min\left\{\frac{L_1 \Delta^{1/2} R}{\varepsilon^{1/2}}, L_1^2 R^2, \frac{L_1 M_0 R^2}{\varepsilon}\right\}$ | Sec. 7.2 and 8.1 (**new**) |
| $(\rho, L_0, L_1)$–Smoothness with $0 \le \rho \le 1$ | $\frac{L_0 R^2 + L_1 M_0^\rho R^2}{\varepsilon}$ | (Li et al., 2024a) |
| | $\frac{L_0 R^2}{\varepsilon} + \min\left\{\frac{L_1 \Delta^{\rho/2} R^{2-\rho}}{\varepsilon^{1-\rho/2}}, \frac{L_1^{2/(2-\rho)} R^2}{\varepsilon^{2(1-\rho)/(2-\rho)}}, \frac{L_1 M_0^\rho R^2}{\varepsilon}\right\}$ | Sec. 7.2 and 8.1 (**new**) |
| $(\rho, L_0, L_1)$–Smoothness with $1 < \rho < 2$ | $\frac{L_0 R^2 + L_1 M_0^\rho R^2}{\varepsilon}$ | (Li et al., 2024a) |
| | $\frac{L_0 R^2}{\varepsilon} + \min\left\{\frac{L_1 \Delta^{\rho/2} R^{2-\rho}}{\varepsilon^{1-\rho/2}}, L_1^{\frac{2}{2-\rho}} R^2 \Delta^{\frac{2(\rho-1)}{2-\rho}}, \frac{L_1 M_0^\rho R^2}{\varepsilon}\right\}$ | Sec. 7.2 and 8.1 (**new**) |
| $(\rho, L_0, L_1)$–Smoothness with $\rho \ge 2$ | $\frac{L_0 R^2 + L_1 M_0^\rho R^2}{\varepsilon}$ | (Li et al., 2024a) |
| | $\frac{L_0 R^2}{\varepsilon} + \min\left\{L_1 \Delta (2M_0)^{\rho-2} + \frac{L_0^{\frac{\rho}{2+\rho}} \Delta^{\frac{\rho}{2+\rho}} L_1^{\frac{2}{2+\rho}} R^{\frac{4}{2+\rho}}}{\varepsilon^{\frac{2}{2+\rho}}}, \frac{L_1 M_0^\rho R^2}{\varepsilon}\right\}$ | Sec. 8.2 (**new**) |
| General Result (works with any $\ell$) | $\frac{\ell(M_0) R^2}{\varepsilon}$ | (Li et al., 2024a) |
| | $\frac{\ell(0) R^2}{\varepsilon} + \min\left\{\bar{T}, \frac{\ell(M_0) R^2}{\varepsilon}\right\}$, where $\bar{T}$ **does not** depend on $\varepsilon$. (convergence rate is $\ell(0) R^2 / \varepsilon$ for $\varepsilon$ small enough) | Sec. 8.3 (**new**) |

[a] The canonical analysis can be found in (Nesterov, 2018; Lan, 2020). However, using non-constant step sizes, it is possible to improve the complexity of GD under $L$–smoothness (Altschuler & Parrilo, 2024; Grimmer et al., 2024).

applicable to virtually any $\ell$ functions.

• In the convex setting, we also improve all known previous results (see Table 2). Specifically, we refine the dominating term[1] from $\frac{\ell(\|\nabla f(x_0)\|) R^2}{\varepsilon}$ (Li et al., 2024a) to $\frac{\ell(0) R^2}{\varepsilon}$ under the $\ell$–smoothness assumptions, which is significant improvement because the initial norm $\|\nabla f(x_0)\|$ can be large. Additionally, we derive tighter non-dominating terms, further enhancing the current theoretical state-of-the-art results (Li et al., 2024a; Gorbunov et al., 2024; Vankov et al., 2024). Even under the well-explored $(L_0, L_1)$–smoothness assumption, we discover a new convergence rate $\frac{L_0 R^2}{\varepsilon} + \min\left\{\frac{L_1 \Delta^{1/2} R}{\varepsilon^{1/2}}, L_1^2 R^2, \frac{L_1 M_0 R^2}{\varepsilon}\right\}$, where the first term in $\min$ is new and can be better in practical regimes (see Section 8).

• We extend our theoretical results to stochastic optimization and verify the obtained results with numerical experiments.

## 2. Motivating Examples

One notable reason for the popularity of the $(L_0, L_1)$–assumption, i.e., $\|\nabla^2 f(x)\| \le L_0 + L_1 \|\nabla f(x)\|$ for all $x \in \mathcal{X}$, is the observation that, in modern neural networks, the spectral norms of the Hessians exhibit a linear dependence on the norm of the gradients (Zhang et al., 2019). However,

there are many examples when $(L_0, L_1)$–assumption fails to hold. The simplest example is $f(x) = -\log x$, which has the Hessian (second derivative) that depends quadratically on the norm of gradient (first derivative). Moreover, we argue that $(L_0, L_1)$–assumption is not appropriate even for modern optimization problems with neural networks. Indeed, consider a toy example $f : \mathbb{R}^2 \to \mathbb{R}$ such that $f(x, y) = \log(1 + \exp(-xy))$. A two-layers neural network with log loss, one feature, and one sample reduces to this function. Then $\nabla f(x, y) = -\frac{1}{1+e^{xy}}(y, x)^\top \in \mathbb{R}^2$ and $\nabla^2 f(x, y) = (\frac{e^{xy}}{(1+e^{xy})^2} y^2, \frac{e^{xy}}{(1+e^{xy})^2} xy - \frac{1}{1+e^{xy}}; \frac{e^{xy}}{(1+e^{xy})^2} xy - \frac{1}{1+e^{xy}}, \frac{e^{xy}}{(1+e^{xy})^2} x^2) \in \mathbb{R}^{2\times 2}$. In the regime $xy = -1$, when the sample is not correctly classified, we get $\nabla f(x, y) \approx -(y, x)^\top$ and $\nabla^2 f(x, y) \approx (y^2, -1; -1, x^2) \in \mathbb{R}^{2\times 2}$. Notice that $\|\nabla^2 f(x, y)\| \approx y^2$ and $\|\nabla f(x, y)\| \approx y$ if $y \to \infty$ and $x = 1/y$, Thus, a more appropriate assumption would be $(\rho, L_0, L_1)$–smoothness, i.e., $\|\nabla^2 f(x, y)\| \le L_0 + L_1 \|\nabla f(x, y)\|^\rho$ for all $x \in \mathcal{X}$ with $\rho \ge 2$. Furthermore, there exist examples of functions that satisfy $(\rho, L_0, L_1)$–smoothness with only $\rho > 2$. For instance, take $f(x) = -\sqrt{1-x}$, which is $(3, L_0, L_1)$–smooth. This is why exploring a more general assumption, $\ell$-smoothness, is important. Another example where the $(L_0, L_1)$–assumption is not satisfied in practice is given in (Cooper, 2024; Chen et al., 2023).

---
[1] terms that dominate when $\varepsilon$ is small.

---

**Algorithm 1** Gradient Descent (GD) with $\ell$-Smoothness

---

1: **Input:** starting point $x_0 \in \mathcal{X}$, function $\ell$ from Assumption 3.1
2: **for** $k = 0, 1, \ldots$ **do**
3:     Find the step size

$$\gamma_k = \int_0^1 \frac{dv}{\ell(\|\nabla f(x_k)\| + \|\nabla f(x_k)\| v)}$$

    analytically or numerically (Simpson's rule or Fig. 1)
4:     $x_{k+1} = x_k - \gamma_k \nabla f(x_k)$
5: **end for**

---

## 3. Assumptions in Nonconvex World

Following (Li et al., 2024a), we consider the following assumption:

**Assumption 3.1.** A function $f : \mathbb{R}^d \to \mathbb{R} \cup \{\infty\}$ is $\ell$–smooth if $f$ is twice differentiable on $\mathcal{X}$, $f$ is continuous on the closure of $\mathcal{X}$, and there exists a *non-decreasing positive locally Lipchitz* function $\ell : [0, \infty) \to (0, \infty)$ such that

$$\left\|\nabla^2 f(x)\right\| \leq \ell(\|\nabla f(x)\|) \tag{4}$$

for all $x \in \mathcal{X}$.

This assumption generalizes $L$–smoothness, $(L_0, L_1)$–smoothness, and $(\rho, L_0, L_1)$–smoothness, i.e., $\left\|\nabla^2 f(x)\right\| \leq L_0 + L_1 \|\nabla f(x)\|^\rho$ for all $x \in \mathcal{X}$. In order to introduce Assumption 3.1, we have to assume that $f$ is twice differentiable. Through this paper, we do not calculate Hessians of $f$ in methods, and we only use them in Assumption 3.1. Later, we will prove (6) (Lemma 4.3) that involves only gradients of $f$, and it can taken as an alternative to Assumption 3.1 if $f$ is not twice differential. However, (6) is much less interpretable and intuitive.

We start out work with the nonconvex setting. Thus additionally to Assumption 3.1, we only consider the standard assumption that the function is bounded:

**Assumption 3.2.** There exists $f^* \in \mathbb{R}$ such that $f(x) \geq f^*$ for all $x \in \mathcal{X}$. We define $\Delta := f(x^0) - f^*$, where $x^0$ is a starting point of numerical methods.

**Notations:** $\mathbb{R}_+ := [0, \infty)$; $\mathbb{N} := \{1, 2, \ldots\}$; $\|x\|$ is the output of the standard Euclidean norm for all $x \in \mathbb{R}^d$; $\langle x, y \rangle = \sum_{i=1}^d x_i y_i$ is the standard dot product; $\|A\|$ is the standard spectral norm for all $A \in \mathbb{R}^{d \times d}$; $g = \mathcal{O}(f)$: exist $C > 0$ such that $g(z) \leq C \times f(z)$ for all $z \in \mathcal{Z}$; $g = \Omega(f)$: exist $C > 0$ such that $g(z) \geq C \times f(z)$ for all $z \in \mathcal{Z}$; $g = \Theta(f)$: $g = \mathcal{O}(f)$ and $g = \Omega(f)$; $g = \widetilde{\Theta}(f)$: the same as $g = \Omega(f)$ but up to logarithmic factors; $g \simeq h$: $g$ and $h$ are equal up to a universal constant.

```python
import scipy.integrate as integrate

def find_step_size(ell, norm_grad):
    h = lambda v: 1 / ell(norm_grad * (1 + v))
    return integrate.quad(h, 0, 1)[0]
```

Figure 1: Python function to compute the step sizes $\{\gamma_k\}$ using SciPy (Virtanen et al., 2020).

## 4. Preliminary Theoretical Properties

Before we state our main properties and theorems, we will introduce the $q$–function:

**Definition 4.1** ($q$–function)**.** Let Assumption 3.1 hold. For all $a \geq 0$, we define the function $q : \mathbb{R}_+ \to [0, q_{\max}(a))$ such that

$$q(s; a) = \int_0^s \frac{dv}{\ell(a + v)} \tag{5}$$

where $q_{\max}(a) := \int_0^\infty \frac{dv}{\ell(a+v)} \in (0, \infty]$.

**Proposition 4.2.** *The function $q$ is* invertible, differentiable, positive, and strongly increasing, *and the inverse function $q^{-1} : [0, q_{\max}(a)) \to \mathbb{R}_+$ of $q$ is also* differentiable, positive, and strongly increasing.

The function $q$ is primarily defined because its inverse, $q^{-1}$, plays a key role in the main results, and $q^{-1}$ does not generally have a (nice) closed-form explicit formula.

Under $L$–smoothness, it is known that $\|\nabla f(y) - \nabla f(x)\| \leq L \|y - x\|$ for all $x, y \in \mathcal{X}$. In the first lemma of the proof, we will obtain a generalized bound under Assumption 3.1:

**Lemma 4.3.** *For all $x, y \in \mathcal{X}$ such that $\|y - x\| \in [0, q_{\max})$, if $f$ is $\ell$–smooth (Assumption 3.1), then*

$$\|\nabla f(y) - \nabla f(x)\| \leq q^{-1}(\|y - x\|; \|\nabla f(x)\|), \tag{6}$$

*where $q$ and $q_{\max} \equiv q_{\max}(\|\nabla f(x)\|)$ are defined in Definition 4.1.*

Let us consider some examples. $L$–smoothness: if $\ell(s) = L$, then $q(s; a) = s/L$ and $q^{-1}(z; a) = Lz$; thus $\|\nabla f(y) - \nabla f(x)\| \leq L \|y - x\|$, recovering the classical assumption. $(L_0, L_1)$–smoothness: if $\ell(s) = L_0 + L_1 s$, then $q^{-1}(z; a) = (L_0 + L_1 a)(\exp(L_1 z) - 1)/L_1$, recovering the result from (Vankov et al., 2024). However, (6) works with virtually any other choice of $\ell$. *Remarkably and unexpectedly, it is never necessary to explicitly compute $q^{-1}$ in theory. We will see later that the final results do not depend on $q^{-1}$.*

*Remark* 4.4. An important note is that instead of Assumption 3.1, we can assume (6) for functions that are not twice differentiable. All the subsequent theory remains valid. However, (6) is arguably less intuitive than (4).

The next step is to generalize the inequality

$$f(y) \leq f(x) + \langle \nabla f(x), y - x \rangle + \frac{L\|y-x\|^2}{2} \quad (7)$$

for all $x, y \in \mathcal{X}$, which is true for $L$–smooth functions.

**Lemma 4.5.** *For all $x, y \in \mathcal{X}$ such that $\|y - x\| \in [0, q_{\max}(\|\nabla f(x)\|))$, if $f$ is $\ell$–smooth (Assumption 3.1), then*

$$\begin{aligned} f(y) \leq\; & f(x) + \langle \nabla f(x), y - x \rangle \\ & + \int_0^{\|y-x\|} q^{-1}(\tau; \|\nabla f(x)\|) d\tau \end{aligned} \quad (8)$$

*where $q$ and $q_{\max}(\|\nabla f(x)\|)$ are defined in Definition 4.1.*

Using the same reasoning as with the previous lemma, this bound generalizes the previous bounds for $L$–smoothness and $(L_0, L_1)$–smoothness.

### 4.1. Derivation of the optimal gradient descent rule

Under $L$–smoothness, it is well-known that $y = x - \frac{1}{L}\nabla f(x)$ is the optimal point that minimizes the upper bound in (7). We now aim to extend this reasoning to (8) and determine the "right" GD method under Assumption 3.1. At first glance, it may seem infeasible to find an explicit formula for the optimal step size using (8). However, surprisingly, we can derive it:

**Corollary 4.6.** *For a fixed $x \in \mathcal{X}$, the upper bound*

$$f(x) + \langle \nabla f(x), y - x \rangle + \int_0^{\|y-x\|} q^{-1}(\tau; \|\nabla f(x)\|) d\tau$$

*from (8) is minimized with*

$$y^* = x - \int_0^1 \frac{dv}{\ell(\|\nabla f(x)\| + \|\nabla f(x)\| v)} \nabla f(x) \in \mathcal{X}.$$

*With the optimal $y^*$, the upper bound equals*

$$f(x) - \|\nabla f(x)\|^2 \int_0^1 \frac{1 - v}{\ell(\|\nabla f(x)\| + \|\nabla f(x)\| v)} dv.$$

This corollary follows from Lemma D.1. By leveraging this result, we can immediately propose a new GD method, detailed in Algorithm 1. The algorithm is the standard GD method but with the step size $\gamma_k$. In some cases, such as $L$–smoothness and $(L_0, L_1)$–smoothness, one can easily calculate $\gamma_k$. Indeed, if $\ell(s) = L$, then $\gamma_k = 1/L$. If $\ell(s) = L_0 + L_1 s$, then $\gamma_k = \frac{1}{L_1\|\nabla f(x_k)\|} \log\left(1 + \frac{L_1\|\nabla f(x_k)\|}{L_0 + L_1\|\nabla f(x_k)\|}\right)$, getting the same rule as in (Vankov et al., 2024). Our rule of the step size in Algorithm 1 works with arbitrary $\ell$ function. If it is not possible explicitly, numerical methods like Simpson's rule can be applied instead or SciPy library (Virtanen et al., 2020) (see Figure 1).

*Remark* 4.7. Since $\ell$ is non-decreasing, we can get the following bounds on the optimal step size: $\frac{1}{\ell(2\|\nabla f(x_k)\|)} \leq \gamma_k = \int_0^1 \frac{dv}{\ell(\|\nabla f(x_k)\| + \|\nabla f(x_k)\|v)} \leq \frac{1}{\ell(\|\nabla f(x_k)\|)}$. One can take the step size rule $\bar{\gamma}_k := \frac{1}{\ell(2\|\nabla f(x_k)\|)}$ instead of $\gamma_k$ and avoid the integration, though this approach may result in a less tight final result. For a clean and rigorous theory, it is crucial to work with the optimal step size $\gamma_k$. Notably, in some parts of our proofs, we leverage elegant properties such as $q^{-1}(\gamma_k \|\nabla f(x_k)\|; \|\nabla f(x_k)\|) = \|\nabla f(x_k)\|$, particularly in the arguments surrounding (34) and (24).

Let us take another example and consider $(\rho, L_0, L_1)$–smoothness, $\ell(s) = L_0 + L_1 s^\rho$ for any $p \geq 0$. Then, in view of Remark 4.7, we obtain $(L_0 + 2^\rho L_1 \|\nabla f(x_k)\|^\rho)^{-1} \leq \gamma_k \leq (L_0 + L_1 \|\nabla f(x_k)\|^\rho)^{-1}$; thus, roughly, $\gamma_k \approx (L_0 + L_1 \|\nabla f(x_k)\|^\rho)^{-1} \approx \min\{1/L_0, 1/L_1 \|\nabla f(x_k)\|\}$ if $\rho = 1$, which coincides with the famous clipping rule (Koloskova et al., 2023; Gorbunov et al., 2024; Vankov et al., 2024). However, one can take not only any $\rho \geq 0$, but also any non-decreasing positive locally Lipchitz function $\ell$. The step size rule in Algorithm 1 is universal.

## 5. Convergence Theory in Nonconvex Setting

In the previous section, we derived the optimal step size rule for GD that minimizes the upper bound (8). We are now ready to present the convergence guarantees of this method in the nonconvex setting.

**Theorem 5.1.** *Suppose that Assumptions 3.1 and 3.2 hold. Then Algorithm 1 guarantees that $f(x_{k+1}) \leq f(x_k) - \frac{\gamma_k}{4}\|\nabla f(x_k)\|^2$ for all $k \geq 0$, and*

$$\min_{k \in \{0,\ldots,T-1\}} \frac{\|\nabla f(x_k)\|^2}{\ell(2\|\nabla f(x_k)\|)} \leq \frac{4\Delta}{T} \quad (9)$$

*for all $T \geq 1$.*

Theorem 5.1 is our main result for nonconvex functions. Evidently, (9) does not provide the convergence rate of $\min_{k \in \{0,\ldots,T-1\}} \|\nabla f(x_k)\|^2$. However, if the function $\psi_2(x) := \frac{x^2}{\ell(2x)}$ is strictly increasing, we finally obtain the rate:

**Corollary 5.2.** *In view of Theorem 5.1 and assuming that the function $\psi_2(x) := \frac{x^2}{\ell(2x)}$ is strictly increasing, we get*

$$\min_{k \in \{0,\ldots,T-1\}} \|\nabla f(x_k)\| \leq \psi_2^{-1}\left(\frac{8\Delta}{T}\right) \quad (10)$$

*for Algorithm 1 and for all $T \geq 1$ such that[2] $8\Delta/T \in \mathrm{im}(\psi_2)$.*

If $\psi_2$ is invertible, there is a straightforward strategy to determine an explicit convergence rate: derive $\psi_2^{-1}$ and apply Corollary 5.2. Below are some illustrative examples:

---

[2]if $\psi_2(\infty) = \infty$, then the corollary is true for all $T \geq 1$.

## 5.1. $L$–smoothness

We start with the simplest and straightforward example: $\ell(s) = L$, indicating that $f$ is $L$–smooth. In this case, we can use Corollary 5.2 with $\psi_2(x) = \frac{x^2}{L}$ and get

$$\min_{k\in\{0,\dots,T-1\}} \|\nabla f(x_k)\| \le \sqrt{\frac{4L\Delta}{T}},$$

which is a classical and optimal result (Carmon et al., 2020) (up to a constant factor).

## 5.2. $(L_0, L_1)$–smoothness

For $(L_0, L_1)$–smoothness, we should take $\ell(s) = L_0 + L_1 s$, use Corollary 5.2, and get

$$\min_{k\in\{0,\dots,T-1\}} \|\nabla f(x_k)\| \le \frac{8L_1\Delta}{T} + \sqrt{\frac{4L_0\Delta}{T}}$$

since $\psi_2^{-1}(z) = L_1 z + \sqrt{L_1^2 z^2 + L_0 z} \le 2L_1 z + \sqrt{L_0 z}$. This rate coincides with the results from (Vankov et al., 2024).

## 5.3. $(\rho, L_0, L_1)$–smoothness with $0 \le \rho \le 2$

As far as we know, this result is new. Applying Theorem 5.1 with $\ell(s) = \ell(s) = L_0 + L_1 s^\rho$, we obtain

$$\min_{k\in\{0,\dots,T-1\}} \frac{\|\nabla f(x_k)\|^2}{L_0 + 2^\rho L_1 \|\nabla f(x_k)\|^\rho} \le \frac{4\Delta}{T}. \quad (11)$$

Using the same reasoning as in the previous sections (see details in Section G), one can show that $\min_{k\in\{0,\dots,T-1\}} \|\nabla f(x_k)\|^2 \le \varepsilon$ after at most

$$\max\left\{\frac{8L_0\Delta}{\varepsilon}, \frac{32L_1\Delta}{\varepsilon^{(2-\rho)/2}}\right\} \quad (12)$$

iterations. For $(2, L_0, L_1)$–smoothness, we get $\max\left\{\frac{8L_0\Delta}{\varepsilon}, 32L_1\Delta\right\}$. In contrast, the previous work (Li et al., 2024a) on $\ell$–smoothness does not guarantee any convergence if $\rho = 2$ for their variant of GD. Unlike (Li et al., 2024a), we use non-constant and adaptive step sizes, which enables us to get better convergence guarantees.

# 6. Superquadratic Growth of $\ell$-Function

Corollary 5.2 works only if $\psi_2(x) = \frac{x^2}{\ell(2x)}$ is strictly increasing. If $\ell(2x)$ grows too quickly, Corollary 5.2 cannot be applied. However, using Theorem 5.1, we can still get convergence guarantees, though with the additional assumption that the gradients are bounded:

**Assumption 6.1** (*we assume it only in this section with superquadratic growth of $\ell$-function and nonconvex setting*)**.** A function $f : \mathbb{R}^d \to \mathbb{R} \cup \{\infty\}$ has bounded gradients for some $M \ge 0 : \|\nabla f(x)\| \le M$ for all $x \in \mathcal{X}$.

In general, $\psi_2$ can behave in a highly non-trivial manner. Therefore, each possible $\ell$ function needs to be analyzed individually. If $\ell(s) = L_0 + L_1 s^\rho$ for $\rho > 2$ or $\ell(s) = L_0 + L_1 s^2 e^s$, then function $\psi_2$ first increases and then starts decreasing, and we can apply the following analysis.

## 6.1. Exponential growth of $\ell$

Let us consider an example when $\ell$ grows exponentially[3]: $\ell(s) = L_0 + L_1 s^2 e^s$. Then $\psi_2(x) = \frac{x^2}{L_0 + 4L_1 x^2 e^{2x}} \ge \frac{1}{2}\min\left\{\frac{x^2}{L_0}, \frac{1}{4L_1 e^{2x}}\right\}$. Theorem 5.1 ensures that $\min_{k\in\{0,\dots,T-1\}} \min\left\{\frac{\|\nabla f(x_k)\|^2}{L_0}, \frac{1}{4L_1 e^2 \|\nabla f(x_k)\|}\right\} \le \frac{8\Delta}{T}$. Thus, either $\min_{k\in\{0,\dots,T-1\}} \|\nabla f(x_k)\| \le \sqrt{\frac{8L_0\Delta}{T}}$ or $\max_{k\in\{0,\dots,T-1\}} \|\nabla f(x_k)\|^2 \ge \frac{1}{2}\log\frac{T}{32L_1\Delta}$. Since the gradients are bounded by $M$, we can conclude that the method finds an $\varepsilon$–stationary point after

$$T = \max\left\{\frac{8L_0\Delta}{\varepsilon}, 32L_1\Delta e^{2M}\right\} \quad (13)$$

iterations.

*Remark* 6.2. One can notice that if the gradients are bounded, then Algorithm 1 is not necessary, since $\|\nabla^2 f(x)\| \le \ell(M)$. In this case, it is sufficient to use the classical GD theory with the step size $\gamma = \frac{1}{\ell(M)} = \frac{1}{L_0 + L_1 M^2 e^M}$. However, one would get the iteration complexity $\mathcal{O}\left(\max\left\{\frac{L_0\Delta}{\varepsilon}, \frac{L_1\Delta M^2 e^M}{\varepsilon}\right\}\right)$, which is worse than (13) up to the constant factors, and depends on $\varepsilon$ in the second term. Thus, our new step size rule provably helps even if $\ell$ grows quickly.

## 6.2. $(\rho, L_0, L_1)$–smoothness with $\rho > 2$

Similarly, we can show that the method finds an $\varepsilon$–stationary after

$$\max\left\{\frac{8L_0\Delta}{\varepsilon}, 64L_1\Delta(2M)^{\rho-2}\right\}$$

steps with $(\rho, L_0, L_1)$–smoothness and $\rho > 2$ (see Sec. H).

# 7. Convergence Theory in Convex Setting

We now analyze how Alg. 1 works with convex problems, and use the following standard assumption.

**Assumption 7.1.** A function $f : \mathbb{R}^d \to \mathbb{R} \cup \{\infty\}$ is *convex* and attains the minimum at a (non-unique) $x_* \in \mathbb{R}^d$. We define $R := \|x_0 - x_*\|$, where $x^0$ is a starting point of numerical methods.

Under generalized $\ell$–smoothness, we can prove a convergence rate for convex functions:

---

[3]We multiply the exponent by $s^2$ for convenience

**Theorem 7.2.** *Suppose that Assumptions 3.1 and 7.1 hold. Additionally, the function $\psi_2(x) = \frac{x^2}{\ell(2x)}$ is strictly increasing and $\psi_2(\infty) = \infty$. Then Algorithm 1 guarantees that*

$$\min_{k \in \{0,\dots,T\}} \frac{f(x_k) - f(x_*)}{\ell\left(\frac{2\sqrt{T+1}(f(x_k)-f(x_*))}{\|x_0-x_*\|}\right)} \leq \frac{\|x_0 - x_*\|^2}{T+1}.$$

As with Theorem 5.1, this theorem provides an implicit convergence rate of $f(x_T) - f(x_*)$. Moreover, the theorem offers guarantees only if $\psi_2(x) = \frac{x^2}{\ell(2x)}$ is strictly increasing and $\psi_2(\infty) = \infty$. We now consider some examples.

### 7.1. $L$–smoothness

If $\ell(s) = L$, then $f(x_T) - f(x_*) \leq \frac{L\|x_0-x_*\|^2}{T+1}$ because $\{f(x_k)\}$ is a decreasing sequence (Theorem 5.1). This is the classical rate of GD (Nesterov, 2018).

### 7.2. $(\rho, L_0, L_1)$–smoothness with $0 < \rho < 2$

In Section K, we show that the method finds $\varepsilon$–solution after at most

$$\max\left\{ \frac{2L_0 R^2}{\varepsilon}, \frac{4L_1^{2/(2-\rho)} R^2}{\varepsilon^{2(1-\rho)/(2-\rho)}} \right\}$$

iterations for $\rho \leq 1$. And after at most

$$\max\left\{ \frac{2L_0 R^2}{\varepsilon}, 16 L_1^{\frac{2}{2-\rho}} R^2 \Delta^{\frac{2(\rho-1)}{2-\rho}} \right\} \tag{14}$$

iterations for $1 < \rho < 2$.

## 8. Alternative Convergence Theory in Convex Setting

The main disadvantage of Theorem 7.2 is that it works only if $\psi_2(x) = \frac{x^2}{\ell(2x)}$ is strictly increasing and $\psi_2(\infty) = \infty$. To address this limitation, we introduce a new proof technique for GD from Algorithm 1, which not only works in cases of superquadratic growth of $\ell$ but also guarantees better convergence rates than Theorem 7.2 in certain practical regimes.

**Theorem 8.1.** *Suppose that Assumptions 3.1 and 7.1 hold. Then Algorithm 1 guarantees $f(x_T) - f(x_*) \leq \varepsilon$ after*

$$\inf_{M>0}\left[ \bar{T}(M) + \frac{\ell(2M)\|x_0-x_*\|^2}{2\varepsilon} \right] \tag{15}$$

*iterations, where $\bar{T}(M)$ is the number of iterations required to obtain $\|\nabla f(x_{\bar{T}(M)})\| \leq M$.*

*Remark* 8.2. This convergence rate can be combined with Theorem 7.2. Thus, one can take the minimum of the results from Theorems 8.1 and 7.2.

We complement this theorem with another important result:

**Theorem 8.3.** *Suppose that Assumptions 3.1 and 7.1 hold. Then the sequence $\|\nabla f(x_k)\|$ is decreasing.*

The idea behind this result is as follows: first, we wait for the moment when GD returns a point $x_{\bar{T}(M)}$ such that $\|\nabla f(x_{\bar{T}(M)})\| \leq M$, which takes $\bar{T}(M)$ iterations. After that, GD works in a region where the norm of the Hessians is bounded by $\ell(2M)$, allowing us to apply classical convergence reasoning. The key observation is that this reasoning remains valid for all $M > 0$; therefore, we take the infimum over $M > 0$.

While Theorem 8.1 does not provide the final convergence rate for $f(x_T) - f(x_*)$, we now illustrate the steps that should be taken next. Due to Theorem 8.3, the sequence $\{\|\nabla f(x_k)\|\}$ is decreasing. Using Theorem 5.1, one should should find $\bar{T}(M)$ as a function of $M$, substitute $\bar{T}(M)$ into (15), and minimize the obtained formula over $M > 0$. Let us illustrate this on $(\rho, L_0, L_1)$–smooth functions.

### 8.1. $(\rho, L_0, L_1)$–smoothness with $0 < \rho \leq 2$

According to (12), $\bar{T}(M) = \max\left\{ \frac{8L_0\Delta}{M^2}, \frac{32L_1\Delta}{M^{2-\rho}} \right\}$ if $M < \|\nabla f(x_0)\|$ and $\bar{T}(M) = 0$ if $M \geq \|\nabla f(x_0)\|$. Since $\ell(s) = L_0 + L_1 s^\rho$, we should minimize (15) and consider

$$\simeq \min\left\{ \inf_{M<\|\nabla f(x_0)\|}\left[ \bar{T}(M) + \frac{\ell(2M)R^2}{2\varepsilon} \right], \right.$$
$$\left. \inf_{M\geq\|\nabla f(x_0)\|}\left[ \bar{T}(M) + \frac{\ell(2M)R^2}{2\varepsilon} \right] \right\}$$

$$\simeq \min\left\{ \inf_{M\geq 0}\left[ \max\left\{ \frac{L_0\Delta}{M^2}, \frac{L_1\Delta}{M^{2-\rho}}, \frac{L_0 R^2}{\varepsilon}, \frac{L_1 M^\rho R^2}{\varepsilon} \right\} \right], \right.$$
$$\left. \max\left\{ \frac{L_0 R^2}{\varepsilon}, \frac{L_1\|\nabla f(x_0)\|^\rho R^2}{\varepsilon} \right\} \right\},$$

where, for simplicity, we ignore all universal constants. The term with inf is minimized with $M = \sqrt{\varepsilon\Delta/R^2}$. Thus, we get

$$\max\left\{ \frac{L_0 R^2}{\varepsilon}, \min\left\{ \frac{L_1\Delta^{\rho/2}R^{2-\rho}}{\varepsilon^{1-\rho/2}}, \frac{L_1\|\nabla f(x_0)\|^\rho R^2}{\varepsilon} \right\} \right\}. \tag{16}$$

Unlike (14), (16) is finite when $\rho = 2$. For $\rho = 1$, this setting reduces to the $(L_0, L_1)$–smooth case with the complexity $\tilde{T} := \mathcal{O}\left(\max\left\{ L_0 R^2/\varepsilon, \min\{L_1\Delta^{1/2}R/\varepsilon^{1/2}, L_1\|\nabla f(x_0)\|R^2/\varepsilon\} \right\}\right)$. This complexity can be better than $\mathcal{O}\left(\max\left\{ L_0 R^2/\varepsilon, \min\{L_1^2 R^2, L_1\|\nabla f(x_0)\|R^2/\varepsilon\} \right\}\right)$ and improve the results by Li et al. (2024a); Gorbunov et al. (2024); Vankov et al. (2024). As an example, consider the convex function $f : \mathbb{R} \to \mathbb{R}$ such that $f(x) = -\mu x + e^{L_1 x}$, which is $(L_1\mu, L_1)$–smooth (see Section Q). Taking $x_0 = 0$, we get $f(x_0) - f(x_*) \leq 1$. At the same time, letting $\mu \to 0$, the distance $R = 1/L_1 |\log(\mu/L_1)|$ diverges to infinity, $\mu L_1 R^2/\varepsilon$ converges to zero, and

**Algorithm 2** Stochastic Gradient Descent (SGD) with $\ell$-Smoothness

1: **Input:** starting point $x_0 \in \mathcal{X}$, function $\ell$ from Assumption 3.1, batch size $B$
2: Find the ratio $r = \sup_{s \geq 0} [\ell(2s)/\ell(s)]$
3: **for** $k = 0, 1, \ldots$ **do**
4:     Calculate $g_k = \frac{1}{B} \sum_{j=1}^{B} \nabla f(x_k; \xi_{kj})$   ($\{\xi_{kj}\}$ are i.i.d.)
5:     $\gamma_k = \dfrac{1}{5r} \displaystyle\int_0^1 \dfrac{dv}{\ell\left(\|g_k\| + \|g_k\| v\right)}$
6:     $x_{k+1} = x_k - \gamma_k g_k$
7: **end for**

$\min\{L_1^2 R^2, {}^{L_1 \|\nabla f(x_0)\| R^2}/\varepsilon\}$ can become arbitrarily larger than ${}^{L_1 \Delta^{1/2} R}/\varepsilon^{1/2}$, because the latter scales linearly with $R$.

### 8.2. $(\rho, L_0, L_1)$–smoothness with $\rho > 2$

Theorem 8.1 works even if $\rho > 2$ in $(\rho, L_0, L_1)$–smoothness. In the convex setting, Assumption 6.1 *is not required* since $\|\nabla f(x_k)\| \leq \|\nabla f(x_0)\|$ for all $k \geq 0$. Similarly to Section 6.2, we can conclude that $\left\|\nabla f(x_{\bar{T}(M)})\right\| \leq M$ for $\bar{T}(M) = \max\left\{\frac{8L_0 \Delta}{M^2}, 64 L_1 \Delta (2 \|\nabla f(x_0)\|)^{\rho-2}\right\}$ if $M < \|\nabla f(x_0)\|$, and $\bar{T}(M) = 0$ if $M \geq \|\nabla f(x_0)\|$. Substituting $\bar{T}(M)$ into (15), we obtain

$$
\frac{L_0 R^2}{\varepsilon} + \min \left\{
\begin{aligned}
& L_1 \Delta (2M_0)^{\rho-2} \\
& + \frac{L_0^{\frac{\rho}{2+\rho}} \Delta^{\frac{2}{2+\rho}} L_1^{\frac{2}{2+\rho}} R^{\frac{4}{2+\rho}}}{\varepsilon^{\frac{2}{2+\rho}}}, \frac{L_1 M_0^{\rho} R^2}{\varepsilon}
\end{aligned}
\right\}, \tag{17}
$$

up to universal constant factors, where $M_0 := \|\nabla f(x_0)\|$.

### 8.3. Convergence guarantees for small $\varepsilon$

In many practical problems, finding a term in convergence rates that dominates when $\varepsilon$ is small is sufficient. We notice that the term ${}^{L_0 R^2}/\varepsilon$ dominates in all derived complexities. It turns out we can generalize this observation:

**Corollary 8.4** (Subquadratic and Quadratic Growth of $\ell$). *Consider Theorem 8.1. Additionally, assume that the function $\psi_2(x) = \frac{x^2}{\ell(2x)}$ is strictly increasing. Then GD finds an $\varepsilon$–solution after at most*

$$
\frac{\ell(0) R^2}{\varepsilon} + \min\left\{\bar{T}(\ell, \Delta), \frac{\ell(2M_0) R^2}{2\varepsilon}\right\} \tag{18}
$$

*iterations, where $M_0 := \|\nabla f(x_0)\|$. $\bar{T}(\ell, \Delta)$ depends only on $\ell$ and $\Delta$, and does not depend on $\varepsilon$.*

**Corollary 8.5** (Superquadratic Growth of $\ell$). *Consider Theorem 8.1. Then GD finds an $\varepsilon$–solution after at most*

$$
\frac{\ell(0) R^2}{\varepsilon} + \min\left\{\bar{T}(\ell, \Delta, M_0), \frac{\ell(2M_0) R^2}{2\varepsilon}\right\} \tag{19}
$$

*iterations, where $M_0 := \|\nabla f(x_0)\|$. $\bar{T}(\ell, \Delta, M_0)$ depends only on $\ell$, $\Delta$, and $M_0$, and does not depend on $\varepsilon$.*

Thus, for small $\varepsilon$, Algorithm 1 converges after at most $\Theta\left({}^{\ell(0) R^2}/\varepsilon\right)$ iterations. In other words, Algorithm 1 behaves like the classical GD method with the step size $1/\ell(0)$. In contrast, Li et al. (2024a) proved a significantly weaker convergence rate $\Theta\left({}^{\ell(\|\nabla f(x_0)\|) R^2}/\varepsilon\right)$ for GD.

## 9. Stochastic Gradient Descent

We can obtain similar results in a stochastic setting, where we access to stochastic gradients $\nabla f(x; \xi)$ characterized by the following "light-tail" assumption (Lan, 2020).

**Assumption 9.1.** For all $x \in \mathcal{X}$, the stochastic gradients $\nabla f(x; \xi)$ satisfy $\mathbb{E}_\xi [\nabla f(x; \xi)] = \nabla f(x)$ and $\mathbb{E}_\xi \left[\exp\left(\|\nabla f(x; \xi) - \nabla f(x)\|^2 / \sigma^2\right)\right] \leq \exp(1)$ for some $\sigma > 0$.

We can prove the following result that extend Theorem 5.1:

**Theorem 9.2.** *Suppose that Assumptions 3.1, 3.2, and 9.1 hold. Let $T$ denote the number required to ensure that $\min_{k \in \{0,\ldots,T-1\}} \|\nabla f(x_k)\|^2 \leq \varepsilon$ based on*

$$
\min_{k \in \{0,\ldots,T-1\}} \frac{(\frac{3}{2} \|\nabla f(x_k)\|)^2}{\ell(3 \|\nabla f(x_k)\|)} \leq r \times \frac{45\Delta}{T}. \tag{20}
$$

*Then, with probability $1 - \delta$ and batch size $B = \max\left\{\left\lceil 32 \left(1 + \sqrt{3 \log(T/\delta)}\right)^2 \sigma^2/\varepsilon \right\rceil, 1\right\}$, Algorithm 2 finds an $\varepsilon$–stationary point after $T$ iterations, and the total number of computed stochastic gradients is $B \times T$.*

Let us explain how the theorem works. Notice that the convergence guarantee (20) coincides with (9), differing only in $r$ and universal constants. To find $T$, one can use exactly the same reasoning as in Section 5. Specifically, $T$ is shown to be the same as in that section, multiplied by $r$ and a universal constant. The main difference lies in $r$. For instance, $r = 2^\rho$ for $(\rho, L_0, L_1)$–smoothness. If $\rho \leq 2$, then $r \leq 4$, meaning it is simply a constant factor. Overall, for any $\ell$, the total number of computed stochastic gradients is $\Theta(B \times T)$. For $(\rho, L_0, L_1)$–smoothness with $0 < \rho \leq 2$, we get

$$
\widetilde{\Theta}\left(\frac{\sigma^2 L_0 \Delta}{\varepsilon^2} + \frac{\sigma^2 L_1 \Delta}{\varepsilon^{(4-\rho)/2}} + \frac{L_0 \Delta}{\varepsilon} + \frac{L_1 \Delta}{\varepsilon^{(2-\rho)/2}}\right),
$$

ignoring the logarithmic factor. For $\varepsilon$ small enough, the dominating term is $\widetilde{\Theta}\left(\sigma^2 L_0 \Delta/\varepsilon^2\right)$, that, up to logarithmic factors, recovers the optimal rate (Arjevani et al., 2022). Furthermore, unlike (Li et al., 2024a), we can extend these convergence guarantees to cases where $\rho \geq 2$, and the dominating term does not depend on $L_1$. Our approach assumes stochastic gradients with "light tails," which is

necessary for handling the non-constant, gradient-dependent step sizes in Algorithm 2. Extending these results to settings with "heavy tails" is an important and challenging research question.

# 10. Conclusion and Future Work

This work offers new insights into modern optimization within the framework of the $\ell$-smoothness assumption. We present new lemmas, algorithms, and convergence rates. However, numerous other directions remain to be explored, including stochastic optimization with "heavy tails" (Robbins & Monro, 1951; Lan, 2020), acceleration of Algorithm 1 (addressing the exponential dependence on $L_1$ and $R$, or resolving the need to solve an auxiliary one-dimensional optimization problem in each iteration (Gorbunov et al., 2024; Vankov et al., 2024)), variance reduction (Schmidt et al., 2017), federated learning, and distributed optimization (Konečný et al., 2016).

# Acknowledgements

The work was supported by the grant for research centers in the field of AI provided by the Ministry of Economic Development of the Russian Federation in accordance with the agreement 000000C313925P4F0002 and the agreement with Skoltech №139-10-2025-033.

# Impact Statement

This paper presents work whose goal is to advance the field of Machine Learning. There are many potential societal consequences of our work, none which we feel must be specifically highlighted here.

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

# A. Experiments

We verify our theoretical results by asking whether it is necessary to use the step size rule from Algorithm 1, and maybe it is sufficient to use the step size rules by Li et al. (2024a); Vankov et al. (2024) instead. We first take the function $f : [0, 0.1] \to \mathbb{R} \cup \{\infty\}$ defined as $f(x) = -\log x - \log(0.1 - x)$ (see Figure 2), which has its minimum at $x_* = 0.05$. This function is $(\rho, 800, 2)$–smooth with $\rho = 2$; however, it is not $(L_0, L_1)$–smooth for any $L_0, L_1 \geq 0$. Consequently, we run Algorithm 1 with $\ell(s) = 800 + 2s^2$, starting at $x_0 = 10^{-7}$, and observe that it converges[4] after 75 iterations. Next, we take the step size $\gamma_k = 1/(800 + 2(2f'(x_0))^2)$ from (Li et al., 2024a) and observe that GD requires at least 20.000 iterations because $f'(x_0)$ is huge. Finally, to verify whether the exponent 2 is necessary, we take $\ell(s) = 800 + 2s$ (Vankov et al., 2024). For this choice of $\ell$, GD diverges.

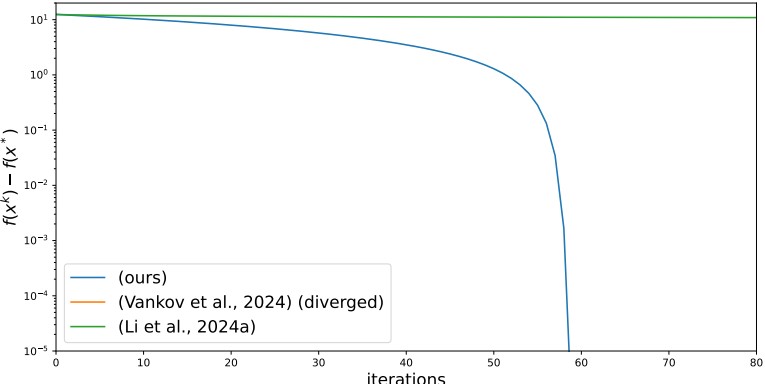

Figure 2: Experiment with $f(x) = -\log x - \log(0.1 - x)$.

We repeat this experiment with the function $f : \mathbb{R} \to \mathbb{R}$ defined as $f(x) = e^x + e^{1-x}$ (see Figure 3) which has its minimum at $x_* = 0.5$. This function is $(3.3, 1)$–smooth, meaning we can run Algorithm 1 with $\ell(s) = 3.3 + s$. It converges after at most 20 iterations. At the same time, if we choose $\ell(s) = 3.3 + s^2$ or $\gamma_k = (3.3 + 2|f'(x_0)|)^{-1}$ (Li et al., 2024a), GD requires at least 200 iterations to converge. These experiments underscore the importance of our step size rule and the right choice of step size and normalization.

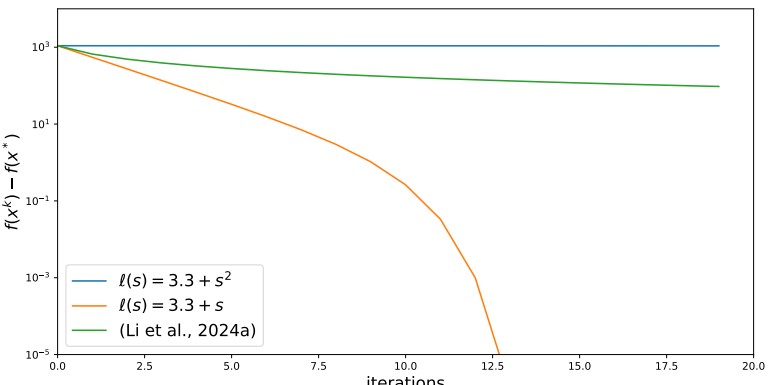

Figure 3: Experiment with $f(x) = e^x + e^{1-x}$.

---

[4]finds $\bar{x}$ such that $f(\bar{x}) - f(x_*) \leq 10^{-5}$

## B. Proof of Lemma 4.3

The following proof is based on the techniques from (Li et al., 2024a; Vankov et al., 2024).

*Proof.* It is sufficient to prove that for all $x, x + th \in \mathcal{X}$, $t \in [0, q_{\max})$ such that $\|h\| = 1$, if $f$ is $\ell$–smooth (Assumption 3.1), then

$$\|\nabla f(x + th) - \nabla f(x)\| \le q^{-1}(t; \|\nabla f(x)\|), \tag{21}$$

where $q$ and $q_{\max} \equiv q_{\max}(\|\nabla f(x)\|)$ are defined in Definition 4.1. One can get (6) from (21) with $y = x + th$ and $t = \|y - x\|$.

Let us define $v(t) := \int_0^t \ell(\|\nabla f(x + h\tau)\|) d\tau$. Using Taylor expansion, we get

$$
\begin{aligned}
\|\nabla f(x + ht) - \nabla f(x)\| &= \left\| \int_0^t \nabla^2 f(x + h\tau) d\tau h \right\| \le \|h\| \int_0^t \left\| \nabla^2 f(x + h\tau) \right\| d\tau \\
&= \int_0^t \left\| \nabla^2 f(x + h\tau) \right\| d\tau \le \int_0^t \ell(\|\nabla f(x + h\tau)\|) d\tau = v(t).
\end{aligned}
\tag{22}
$$

In the first inequality, we use the definition of the operator norm and the triangle inequality for the integral. The triangle inequality yields

$$\|\nabla f(x + ht)\| \le \|\nabla f(x)\| + \|\nabla f(x + ht) - \nabla f(x)\| \le \|\nabla f(x)\| + v(t).$$

Note that $v'(t) = \ell(\|\nabla f(x + ht)\|)$ and $\ell$ is non-decreasing. Thus $v'(t) \le \ell(\|\nabla f(x)\| + v(t))$ for all $t \ge 0$ and $v(0) = 0$. Instead of this inequality, consider the differential equation

$$g'(t) = \ell(\|\nabla f(x)\| + g(t)), \quad g(0) = 0, \tag{23}$$

where $g : \mathbb{R}_+ \to \mathbb{R}$ is a solution, and $\|\nabla f(x)\|$ is a fixed quantity. Using the standard differential algebra, we can solve it:

$$\frac{dg(t)}{\ell(\|\nabla f(x)\| + g(t))} = dt \Rightarrow \int_0^t \frac{dg(v)}{\ell(\|\nabla f(x)\| + g(v))} = t \Rightarrow \int_0^{g(t)} \frac{dv}{\ell(\|\nabla f(x)\| + v)} = t.$$

Recall the definition of the function $q$, which is an increasing and differentiable function on $\mathbb{R}_+$ with $q'(s; \cdot, \cdot) > 0$ for all $s \in \mathbb{R}_+$. Therefore, $q^{-1}$ is strongly increasing and differentiable on $[0, q_{\max})$ and we can take $g(t) = q^{-1}(t; \|\nabla f(x)\|)$ for all $t \in [0, q_{\max})$. One can check that $g(t)$ is a solution of (23). Let us define $p(t) := v(t) - g(t)$. Then $p(0) = 0$ and

$$p'(t) = (v(t) - g(t))' \le \ell(\|\nabla f(x)\| + v(t)) - \ell(\|\nabla f(x)\| + g(t)).$$

Let us fix any $b \in [0, q_{\max})$. Since $\ell$ is locally Lipchitz, there exists $M \ge 0$ such that

$$p'(t) \le M \|h\| (v(t) - g(t)) = M \|h\| p(t)$$

for all $t \in [0, b]$ because $v(t)$ and $g(t)$ are bounded on $[0, b]$. Using Grönwall's lemma (Gronwall, 1919), we can conclude that $p(t) \le p(0) \exp(Mt) = 0$. Thus $v(t) \le g(t) = q^{-1}(t; \|\nabla f(x)\|)$ for all $t \in [0, b]$. The last inequality holds for all $t \in [0, q_{\max})$ because $b$ is an arbitrary value from $[0, q_{\max})$. Finally, using (22), we get (21). $\square$

## C. Proof of Lemma 4.5

*Proof.* If $x = y$, then the lemma holds, since $q^{-1}$ is non-negative. Otherwise, using Taylor expansion, we get

$$
\begin{aligned}
f(y) &= f(x) + \langle \nabla f(x), y - x \rangle + \int_0^1 \langle \nabla f(x + \tau(y - x)) - \nabla f(x), y - x \rangle d\tau \\
&\le f(x) + \langle \nabla f(x), y - x \rangle + \|y - x\| \int_0^1 \|\nabla f(x + \tau(y - x)) - \nabla f(x)\| d\tau.
\end{aligned}
$$

Due to Lemma 4.3:

$$f(y) \le f(x) + \langle \nabla f(x), y - x \rangle + \|y - x\| \int_0^1 q^{-1}(\tau \|y - x\| ; \|\nabla f(x)\|) d\tau$$

$$= f(x) + \langle \nabla f(x), y - x \rangle + \int_0^{\|y-x\|} q^{-1}(\tau; \|\nabla f(x)\|) d\tau.$$

where we changed variables in the integral. □

## D. Auxiliary Lemmas

We need this auxiliary lemma to find the optimal choice of $\gamma_k$ in Algorithm 1, and in Lemma I.1.

**Lemma D.1.** *For all $z \in \mathbb{R}^d$ and $s \geq 0$, the term*

$$U \equiv U(t, h) := \langle z, th \rangle + \int_0^t q^{-1}(\tau; s) d\tau$$

*under the constraints $\|h\| = 1$ and $t \in [0, q_{\max}(s))$ is minimized with $t^* = q(\|z\|; s) = \|z\| \int_0^1 \frac{dv}{\ell(s+\|z\|v)}$ and $h^* = -\frac{z}{\|z\|}$ if $z \neq 0$, and with $t^* = 0$ and any $h^* \in \mathbb{R}^d$ such that $\|h^*\| = 1$ if $z = 0$. With the optimal $t^*$ and $h^*$, the term $U$ equals*

$$U^* := -\|z\|^2 \int_0^1 \frac{1 - v}{\ell(s + \|z\| v)} dv.$$

*Proof.* The term $U$ depends on $h$ only in $\langle z, th \rangle$. Since $\|h\| = 1$, the term is minimized with $h^* = -\frac{z}{\|z\|}$ if $z \neq 0$, and with any $h^* \in \mathbb{R}^d$ such that $\|h^*\| = 1$ if $z = 0$. In both cases, we get

$$U = -t \|z\| + \int_0^t q^{-1}(\tau; s) d\tau$$

with an optimal choice of $h$. Next, we can take the derivative w.r.t. $t$ and obtain

$$U_t' = -\|z\| + q^{-1}(t; s).$$

$U_t'$ is strongly increasing because $q^{-1}$ is strongly increasing (see Definition 4.1). Moreover $U_t' \leq 0$ when $t = 0$. Thus, there exists the optimal $t^*$ defined by the equation

$$-\|z\| + q^{-1}(t^*; s) = 0 \Leftrightarrow t^* = q(\|z\|; s). \tag{24}$$

If $\|z\| = 0$, then $t^* = 0$. Otherwise,

$$t^* = \int_0^{\|z\|} \frac{dv}{\ell(s + v)} = \|z\| \int_0^1 \frac{dv}{\ell(s + \|z\| v)}$$

For this choice of $t^*$ and $h^*$, using the change of variables $p = q^{-1}(\tau; s)$, we get

$$U = -q(\|z\|; s) \|z\| + \int_0^{q(\|z\|; s)} q^{-1}(\tau; s) d\tau = -q(\|z\|; s) \|z\| + \int_0^{\|z\|} p \, dq(p; s)$$

$$\overset{(5)}{=} -\|z\| \int_0^{\|z\|} \frac{1}{\ell(s + v)} dv + \int_0^{\|z\|} \frac{v}{\ell(s + v)} dv = -\|z\|^2 \int_0^1 \frac{1 - v}{\ell(s + \|z\| v)} dv.$$

□

We will require the following lemma to ensure that Algorithm 1 is well-defined. This lemma says we can safely do a step in a direction with a step size $t \in [0, q_{\max}(\|\nabla f(x)\|))$.

**Lemma D.2.** *For a fixed $x \in \mathcal{X}$, the point $y = x + th \in \mathcal{X}$ for all $t \in [0, q_{\max}(\|\nabla f(x)\|))$ and $h \in \mathbb{R}^d$ such that $\|h\| = 1$.*

The proof is a bit technical and can be skipped by the reader if $\mathcal{X} = \mathbb{R}^d$.

*Proof.* Let us define $y(\mu) := x + \mu t h$ for all $\mu \in [0, 1]$. Note that $y(1) = y$ and $y(0) = x \in \mathcal{X}$. Using proof by contradiction, assume that $y(1) \notin \mathcal{X}$. Then

$$\bar{\mu} := \sup\{\mu \geq 0 \,|\, y(\mu) \in \mathcal{X}\} \in [0, 1],$$

$y(\bar{\mu}) \notin \mathcal{X}$, and $y(\bar{\mu})$ belongs to the closure of $\mathcal{X}$ because $\mathcal{X}$ is open convex and $f$ is continuous on the closure of $\mathcal{X}$. Next, $y(\mu) \in \mathcal{X}$ for all $\mu < \bar{\mu}$ and

$$f(y(\mu)) \leq f(x) + \mu t \langle \nabla f(x), h \rangle + \int_0^{\mu t} q^{-1}(\tau; \|\nabla f(x)\|) d\tau \tag{25}$$

for all $\mu < \bar{\mu}$ due to Lemma 4.5. Since the r.h.s. is continuous for all $\mu \in [0, 1]$ because $t \in [0, q_{\max}(\|\nabla f(x)\|))$, we can take a sequence $\{\mu_n\}$ such that $\mu_n < \bar{\mu}$ and $\lim_{n \to \infty} \mu_n = \bar{\mu}$. Thus

$$\lim_{n \to \infty} \left[ f(x) - \mu_n t \langle \nabla f(x), h \rangle + \int_0^{\mu_n t} q^{-1}(\tau; \|\nabla f(x)\|) d\tau \right]$$
$$= f(x) - \bar{\mu} t \langle \nabla f(x), h \rangle + \int_0^{\bar{\mu} t} q^{-1}(\tau; \|\nabla f(x)\|) d\tau < \infty. \tag{26}$$

Since $f$ is continuous on the closure of $\mathcal{X}$, we have

$$f(y(\bar{\mu})) = \lim_{n \to \infty} f(y(\mu_n)) \overset{(25),(26)}{<} \infty,$$

which contradicts $y(\bar{\mu}) \notin \mathcal{X}$. $\qquad \square$

## E. Proof of Corollary 4.6

*Proof.* Corollary 4.6 follows from Lemma D.1 with $s = \|\nabla f(x)\|$, $z = \nabla f(x)$, and $y^* = x + t^* h^*$. It is only left to show that $y^* \in \mathcal{X}$, which follows from Lemma D.2. $\qquad \square$

## F. Proof of Theorem 5.1

*Proof.* Notice that Algorithm 1 uses the optimal gradient descent rule from Corollary 4.6. Thus, for Algorithm 1, we have

$$f(x_{k+1}) \leq f(x_k) - \|\nabla f(x_k)\|^2 \int_0^1 \frac{1 - v}{\ell(\|\nabla f(x_k)\| + \|\nabla f(x_k)\| v)} dv. \tag{27}$$

Corollary 4.6 automatically ensures that $x_{k+1} \in \mathcal{X}$ if $x_k \in \mathcal{X}$. Thus Algorithm 1 is well-defined because $x_0 \in \mathcal{X}$. Note that

$$\int_0^1 \frac{1 - v}{\ell(\|\nabla f(x_k)\| + \|\nabla f(x_k)\| v)} dv \geq \int_0^{1/2} \frac{1 - v}{\ell(\|\nabla f(x_k)\| + \|\nabla f(x_k)\| v)} dv$$
$$\geq \frac{1}{2} \int_0^{1/2} \frac{1}{\ell(\|\nabla f(x_k)\| + \|\nabla f(x_k)\| v)} dv \geq \frac{1}{4} \int_0^1 \frac{1}{\ell(\|\nabla f(x_k)\| + \|\nabla f(x_k)\| v)} dv = \frac{\gamma_k}{4}.$$

where the last inequality because $\ell$ is non-decreasing. Substituting this to (27), we get

$$f(x_{k+1}) \leq f(x_k) - \frac{\gamma_k}{4} \|\nabla f(x_k)\|^2.$$

Summing this inequality for $k = 0, \ldots, T - 1$, dividing the result by $T$, taking the minimum over $k \in \{0, \ldots, T - 1\}$, and using Assumption 3.2 and Remark 4.7, one can get (9). $\qquad \square$

## G. Derivation of the Rate from Section 5.3

Let us now apply Theorem 5.1 with $\ell(s) = \ell(s) = L_0 + L_1 s^\rho$ and $0 \leq \rho \leq 2$:

$$\min_{k \in \{0, \ldots, T-1\}} \frac{\|\nabla f(x_k)\|^2}{L_0 + 2^\rho L_1 \|\nabla f(x_k)\|^\rho} \leq \frac{4\Delta}{T}. \tag{28}$$

*Remark* G.1. Due to Theorem 5.1 and Remark 4.7, the term $2^\rho$ is not tight and could potentially be improved. However, achieving this improvement would require calculating the integral in $\gamma_k$ explicitly instead, which results in "lengthy trigonometric formulas."

We continue with (28) to get

$$\min_{k\in\{0,\dots,T-1\}} \min\left\{ \frac{\|\nabla f(x_k)\|^2}{L_0}, \frac{\|\nabla f(x_k)\|^{2-\rho}}{2^\rho L_1} \right\} \leq \frac{8\Delta}{T} \tag{29}$$

for all $\rho \geq 0$ because $2\max\{x,y\} \geq x+y$ for all $x,y \geq 0$. If $\rho \leq 2$, then we can guarantee $\min_{k\in\{0,\dots,T-1\}} \|\nabla f(x_k)\|^2 \leq \varepsilon$ after at most

$$\max\left\{ \frac{8L_0\Delta}{\varepsilon}, \frac{32L_1\Delta}{\varepsilon^{(2-\rho)/2}} \right\}$$

iterations.

## H. Derivation of the Rate from Section 6.2

Using (29), either

$$\min_{k\in\{0,\dots,T-1\}} \|\nabla f(x_k)\|^2 \leq \frac{8L_0\Delta}{T} \quad \text{or} \quad \max_{k\in\{0,\dots,T-1\}} \|\nabla f(x_k)\|^{\rho-2} \geq \frac{T}{2^{\rho+3}L_1\Delta}.$$

We now require Assumption 6.1. Since the gradients are bounded by $M$, we can conclude that the method finds an $\varepsilon$–stationary after

$$\max\left\{ \frac{8L_0\Delta}{\varepsilon}, 64L_1\Delta(2M)^{\rho-2} \right\}$$

iterations because $\max_{k\in\{0,\dots,T-1\}} \|\nabla f(x_k)\|^{\rho-2} \leq M^{\rho-2}$.

## I. Proof of Lemmas in the Convex World

The following proof technique is based on the classical approaches by Nesterov (2018). We start our proof by generalizing the inequalities $\frac{1}{2L}\|\nabla f(x) - \nabla f(y)\|^2 \leq f(x) - f(y) - \langle \nabla f(y), x-y\rangle$ and $\|\nabla f(x) - \nabla f(y)\|^2 \leq L\langle \nabla f(x) - \nabla f(y), x-y\rangle$, which are true under $L$–smoothness (Nesterov, 2018).

**Lemma I.1.** *For all $x,y \in \mathcal{X}$, if $f$ is $\ell$–smooth (Assumption 3.1) and convex, then*

$$\|\nabla f(x) - \nabla f(y)\|^2 \int_0^1 \frac{1-v}{\ell(\|\nabla f(x)\| + \|\nabla f(x) - \nabla f(y)\|v)} dv \leq f(x) - f(y) - \langle \nabla f(y), x-y\rangle \tag{30}$$

*and*

$$\begin{aligned}
\langle \nabla f(x) &- \nabla f(y), x-y\rangle \\
&\geq \|\nabla f(x) - \nabla f(y)\|^2 \int_0^1 \left( \frac{1-v}{\ell(\|\nabla f(x)\| + \|\nabla f(x) - \nabla f(y)\|v)} + \frac{1-v}{\ell(\|\nabla f(y)\| + \|\nabla f(x) - \nabla f(y)\|v)} \right) dv.
\end{aligned} \tag{31}$$

*Proof.* As in (Nesterov, 2018), we define the function $\phi(y) := f(y) - \langle \nabla f(x_0), y\rangle$ for a fixed $x_0 \in \mathcal{X}$. For all $x \in \mathcal{X}$, we have

$$\begin{aligned}
\phi(x_0) = \min_{y\in\mathcal{X}} \phi(y) &= \min_{y\in\mathcal{X}} \{f(y) - \langle \nabla f(x_0), y\rangle\} \\
&\overset{\text{Lemma 4.5}}{\leq} \min_{y\in\mathcal{X}} \left\{ f(x) + \langle \nabla f(x), y-x\rangle + \int_0^{\|y-x\|} q^{-1}(\tau; \|\nabla f(x)\|)d\tau - \langle \nabla f(x_0), y\rangle \right\}
\end{aligned}$$

$$= \min_{y \in \mathcal{X}} \left\{ f(x) + \langle \nabla f(x) - \nabla f(x_0), y - x \rangle + \int_0^{\|y-x\|} q^{-1}(\tau; \|\nabla f(x)\|) d\tau - \langle \nabla f(x_0), x \rangle \right\}$$

$$= \phi(x) + \min_{y \in \mathcal{X}} \left\{ \langle \nabla f(x) - \nabla f(x_0), y - x \rangle + \int_0^{\|y-x\|} q^{-1}(\tau; \|\nabla f(x)\|) d\tau \right\}$$

$$\leq \phi(x) + \min_{\|h\|=1, t \in [0, q_{\max}(\|\nabla f(x)\|)]} \left\{ \langle \nabla f(x) - \nabla f(x_0), th \rangle + \int_0^t q^{-1}(\tau; \|\nabla f(x)\|) d\tau \right\}.$$

Lemma D.1 with $z = \nabla f(x) - \nabla f(x_0)$ and $s = \|\nabla f(x)\|$ ensures that

$$\phi(x_0) \leq \phi(x) - \|\nabla f(x) - \nabla f(x_0)\|^2 \int_0^1 \frac{1-v}{\ell(\|\nabla f(x)\| + \|\nabla f(x) - \nabla f(x_0)\| v)} dv.$$

Thus, we get (30). One can get (31) interchanging $x$ and $y$ in (30) and using (30) two times. $\square$

**Lemma I.2.** *Suppose that Assumptions 3.1 and 7.1 hold. Then Algorithm 1 guarantees that*

$$\frac{1}{2} \|x_{k+1} - x_*\|^2 \leq \frac{1}{2} \|x_k - x_*\|^2 - \frac{f(x_k) - f(x_*)}{\ell(2 \|\nabla f(x_k)\|)}$$

*for all $k \geq 0$.*

*Proof.* Due to the strategy from Alg. 1, we get

$$\|x_{k+1} - x_*\|^2 = \|x_k - x_*\|^2 - 2\gamma_k \langle x_k - x_*, \nabla f(x_k) \rangle + \gamma_k^2 \|\nabla f(x_k)\|^2. \tag{32}$$

We now consider the last two terms:

$$- 2\gamma_k \langle x_k - x_*, \nabla f(x_k) \rangle + \gamma_k^2 \|\nabla f(x_k)\|^2$$
$$= 2\gamma_k \left( -f(x_*) + f(x_k) + \langle \nabla f(x_k), x_* - x_k \rangle - f(x_k) + f(x_*) \right) + \gamma_k^2 \|\nabla f(x_k)\|^2$$
$$\overset{(30)}{\leq} \gamma_k \|\nabla f(x_k)\|^2 \left( \gamma_k - 2 \int_0^1 \frac{1-v}{\ell(\|\nabla f(x_k)\| v)} dv \right) - 2\gamma_k \left( f(x_k) - f(x_*) \right).$$

Note that

$$\gamma_k - 2 \int_0^1 \frac{1-v}{\ell(\|\nabla f(x_k)\| v)} dv = \int_0^1 \frac{dv}{\ell(\|\nabla f(x_k)\| + \|\nabla f(x_k)\| v)} - 2 \int_0^1 \frac{1-v}{\ell(\|\nabla f(x_k)\| v)} dv$$
$$\leq \int_0^1 \frac{(2v-1)dv}{\ell(\|\nabla f(x_k)\| v)} \leq 0$$

because $\ell$ is non-decreasing. Thus

$$- 2\gamma_k \langle x_k - x_*, \nabla f(x_k) \rangle + \gamma_k^2 \|\nabla f(x_k)\|^2 \leq -2\gamma_k \left( f(x_k) - f(x_*) \right).$$

Substituting this inequality to (32), we get

$$\|x_{k+1} - x_*\|^2 \leq \|x_k - x_*\|^2 - 2\gamma_k \left( f(x_k) - f(x_*) \right)$$

and

$$\frac{1}{2} \|x_{k+1} - x_*\|^2 \leq \frac{1}{2} \|x_k - x_*\|^2 - \frac{f(x_k) - f(x_*)}{\ell(2 \|\nabla f(x_k)\|)}$$

due to Remark 4.7. $\square$

# J. Proof of Theorem 7.2

The following proof is based on the techniques from (Vankov et al., 2024).

*Proof.* Using Lemma I.2, we obtain

$$\frac{1}{2} \|x_{k+1} - x_*\|^2 \leq \frac{1}{2} \|x_k - x_*\|^2 - \frac{f(x_k) - f(x_*)}{\ell(2 \|\nabla f(x_k)\|)}$$

At the same time, using Theorem 5.1 and Remark 4.7, we obtain

$$\frac{\|\nabla f(x_k)\|^2}{4\ell(2 \|\nabla f(x_k)\|)} \leq \frac{\gamma_k}{4} \|\nabla f(x_k)\|^2 \leq f(x_k) - f(x_*)$$

and

$$\frac{\|\nabla f(x_k)\|^2}{\ell(2 \|\nabla f(x_k)\|)} \leq 4(f(x_k) - f(x_*)).$$

Defining $f_k := f(x_k) - f(x_*)$ and $\psi_2(x) := \frac{x^2}{\ell(2x)}$, we get[5] $\|\nabla f(x_k)\|^2 \leq \psi_2^{-1}(4f_k)$ and

$$\frac{1}{2} \|x_{k+1} - x_*\|^2 \leq \frac{1}{2} \|x_k - x_*\|^2 - \frac{f_k}{\ell(2\psi_2^{-1}(4f_k))}$$

$$= \frac{1}{2} \|x_k - x_*\|^2 - \frac{\left(\psi_2^{-1}(4f_k)\right)^2 \times f_k}{\left(\psi_2^{-1}(4f_k)\right)^2 \times \ell(2\psi_2^{-1}(4f_k))}.$$

where the first inequality because $\ell$ is non-decreasing. The equality $\frac{\left(\psi_2^{-1}(4f_k)\right)^2}{\ell(2\psi_2^{-1}(4f_k))} = \psi_2(\psi_2^{-1}(4f_k)) = 4f_k$ simplifies the last inequality to

$$\frac{1}{2} \|x_{k+1} - x_*\|^2 \leq \frac{1}{2} \|x_k - x_*\|^2 - \frac{4(f_k)^2}{\left(\psi_2^{-1}(4f_k)\right)^2}.$$

Summing the inequality for $k \in \{0, \ldots, T-1\}$, we obtain

$$\sum_{k=0}^{T-1} \frac{4(f_k)^2}{\left(\psi_2^{-1}(4f_k)\right)^2} \leq \frac{1}{2} \|x_0 - x_*\|^2,$$

$$\min_{k \in \{0,\ldots,T-1\}} \frac{(f_k)^2}{\left(\psi_2^{-1}(4f_k)\right)^2} \leq \frac{\|x_0 - x_*\|^2}{8T},$$

and

$$\min_{k \in \{0,\ldots,T-1\}} \frac{f_k}{\psi_2^{-1}(4f_k)} \leq \frac{\|x_0 - x_*\|}{2\sqrt{T}}.$$

For all $x, t > 0$, notice that

$$\frac{x}{\psi_2^{-1}(4x)} = t \Leftrightarrow \frac{x}{t} = \psi_2^{-1}(4x) \Leftrightarrow \psi_2\left(\frac{x}{t}\right) = 4x \Leftrightarrow \frac{\frac{x^2}{t^2}}{\ell\left(\frac{x}{t}\right)} = 4x \Leftrightarrow \frac{x}{4\ell\left(\frac{x}{t}\right)} = t^2.$$

Thus

$$\min_{k \in \{0,\ldots,T-1\}} \frac{f_k}{\ell\left(\frac{2\sqrt{T}f_k}{\|x_0 - x_*\|}\right)} \leq \frac{\|x_0 - x_*\|^2}{T}.$$

$\square$

---

[5]This is the place we use that the function $\psi_2(x) = \frac{x^2}{\ell(2x)}$ is strictly increasing and $\psi_2(\infty) = \infty$.

## K. Derivation of the Rate from Section 7.2

Let us define $f_k := f(x_k) - f(x_*)$. Then $f_T = \min_{k \in \{0,\dots,T\}} f_k$ and $\Delta = f_0$. For $(\rho, L_0, L_1)$–smoothness, we should take $\ell(s) = L_0 + L_1 s^\rho$, use Theorem 7.2, and get $\min_{k \in \{0,\dots,T\}} \frac{f_k}{L_0 + L_1 \left(\frac{2\sqrt{T+1}f_k}{R}\right)^\rho} \le \frac{R^2}{T+1}$. Using the inequality $x + y \le 2\max\{x,y\}$, we obtain $\min_{k \in \{0,\dots,T\}} \min\left\{\frac{f_k}{2L_0}, \frac{1}{2L_1 f_k^{\rho-1}\left(\frac{2\sqrt{T+1}}{R}\right)^\rho}\right\} \le \frac{R^2}{T+1}$. For $\rho \le 1$, the method finds $\varepsilon$–solution after at most

$$T := \left\lceil \max\left\{\frac{2L_0 R^2}{\varepsilon}, \frac{16 L_1^{2/(2-\rho)} R^2}{\varepsilon^{2(1-\rho)/(2-\rho)}}\right\}\right\rceil$$

iterations. One can easily show this by swapping $\min_{k \in 0,\dots,T}$ min, noticing that the terms under the min operations are non-decreasing functions of $f_k$ and $f_T = \min_{k \in 0,\dots,T} f_k$, and considering three cases: (i) $\frac{f_T}{2L_0} \le \frac{1}{2L_1 f_T^{\rho-1}\left(\frac{2\sqrt{T+1}}{R}\right)^\rho}$, (ii) $\frac{f_T}{2L_0} > \frac{1}{2L_1 f_T^{\rho-1}\left(\frac{2\sqrt{T+1}}{R}\right)^\rho}$ and $\rho = 1$, and (iii) $\frac{f_T}{2L_0} > \frac{1}{2L_1 f_T^{\rho-1}\left(\frac{2\sqrt{T+1}}{R}\right)^\rho}$ and $\rho < 1$.

For $1 < \rho < 2$, either $f_T \le \frac{2L_0 R^2}{T+1}$ or $\frac{(T+1)^{1-\frac{\rho}{2}}}{L_1 2^{\rho+1} R^{2-\rho}} \le \max_{k \in \{0,\dots,T\}} f_k^{\rho-1}$. Since $f_k$ is decreasing (see Theorem 5.1), the second option implies $(T+1)^{1-\frac{\rho}{2}} \le L_1 2^{\rho+1} R^{2-\rho} f_0^{\rho-1}$, meaning the method finds $\varepsilon$–solution after at most

$$\max\left\{\frac{2L_0 R^2}{\varepsilon}, 16 L_1^{\frac{2}{2-\rho}} R^2 \Delta^{\frac{2(\rho-1)}{2-\rho}}\right\}$$

iterations.

## L. Proof of Theorem 8.1

*Proof.* Using Lemma I.2, we obtain

$$\frac{1}{2}\|x_{k+1} - x_*\|^2 \le \frac{1}{2}\|x_k - x_*\|^2 - \frac{f(x_k) - f(x_*)}{\ell(2\|\nabla f(x_k)\|)} \tag{33}$$

for all $k \ge 0$. $\|\nabla f(x_k)\|$ is decreasing due to Theorem 8.3. We fix any $\bar{T} \ge 0$. Thus, for all $k \ge \bar{T}$, we get

$$\frac{1}{2}\|x_{k+1} - x_*\|^2 \le \frac{1}{2}\|x_k - x_*\|^2 - \frac{f(x_k) - f(x_*)}{\ell(2\|\nabla f(x_{\bar{T}})\|)}$$

Summing the last inequality for $k = \bar{T}, \dots, T$ and dividing by $T - \bar{T} - 1$, we obtain

$$f(x_T) - f(x_*) \le \frac{\ell(2\|\nabla f(x_{\bar{T}})\|)\|x_{\bar{T}} - x_*\|^2}{2(T - \bar{T} + 1)} \le \frac{\ell(2\|\nabla f(x_{\bar{T}})\|)\|x_0 - x_*\|^2}{2(T - \bar{T} + 1)},$$

where use the inequalities $\|x_0 - x_*\|^2 \ge \|x_{\bar{T}} - x_*\|^2 \ge \|x_T - x_*\|^2 \ge 0$ due to (33). For any $M \ge 0$, taking $\bar{T}(M)$ such that $\|\nabla f(x_{\bar{T}})\| \le M$, we get

$$f(x_T) - f(x_*) \le \frac{\ell(2M)\|x_0 - x_*\|^2}{2(T - \bar{T}(M) + 1)}.$$

Thus, after

$$\bar{T}(M) + \frac{\ell(2M)\|x_0 - x_*\|^2}{2\varepsilon}$$

iterations the inequality $f(x_T) - f(x_*) \le \varepsilon$ holds. The final result holds since $M > 0$ is arbitrary. $\qquad\square$

## M. Proof of Theorem 8.3

*Proof.* We have

$$\|\nabla f(x_{k+1})\|^2 = \|\nabla f(x_k)\|^2 + 2\langle \nabla f(x_k), \nabla f(x_{k+1}) - \nabla f(x_k)\rangle + \|\nabla f(x_{k+1}) - \nabla f(x_k)\|^2$$
$$= \|\nabla f(x_k)\|^2 - \frac{2}{\gamma_k}\langle \nabla f(x_{k+1}) - \nabla f(x_k), x_{k+1} - x_k\rangle + \|\nabla f(x_{k+1}) - \nabla f(x_k)\|^2.$$

Lemma 4.3 guarantees that

$$\|\nabla f(x_{k+1}) - \nabla f(x_k)\| \leq q^{-1}(\gamma_k \|\nabla f(x_k)\|\,;\|\nabla f(x_k)\|).$$

Notice that

$$\gamma_k \|\nabla f(x_k)\| = \int_0^1 \frac{d\,\|\nabla f(x_k)\|\,v}{\ell(\|\nabla f(x_k)\| + \|\nabla f(x_k)\|\,v)} = \int_0^{\|\nabla f(x_k)\|} \frac{dv}{\ell(\|\nabla f(x_k)\| + v)} \overset{\text{Def.5}}{=} q(\|\nabla f(x_k)\|\,;\|\nabla f(x_k)\|).$$

Thus $q^{-1}(\gamma_k \|\nabla f(x_k)\|\,;\|\nabla f(x_k)\|) = \|\nabla f(x_k)\|$ and

$$\|\nabla f(x_{k+1}) - \nabla f(x_k)\| \leq \|\nabla f(x_k)\|. \tag{34}$$

Ignoring the first non-negative term in the integral of (31), we obtain

$$\langle \nabla f(x_{k+1}) - \nabla f(x_k), x_{k+1} - x_k\rangle \geq \|\nabla f(x_{k+1}) - \nabla f(x_k)\|^2 \int_0^1 \frac{1-v}{\ell(\|\nabla f(x_k)\| + \|\nabla f(x_{k+1}) - \nabla f(x_k)\|\,v)}dv$$
$$\overset{(34)}{\geq} \|\nabla f(x_{k+1}) - \nabla f(x_k)\|^2 \int_0^1 \frac{1-v}{\ell(\|\nabla f(x_k)\| + \|\nabla f(x_k)\|\,v)}dv.$$

Therefore

$$\|\nabla f(x_{k+1})\|^2 \leq \|\nabla f(x_k)\|^2 - \frac{1}{\gamma_k}\left(2\int_0^1 \frac{1-v}{\ell(\|\nabla f(x_k)\| + \|\nabla f(x_k)\|\,v)}dv - \gamma_k\right)\|\nabla f(x_{k+1}) - \nabla f(x_k)\|^2$$
$$= \|\nabla f(x_k)\|^2 - \frac{1}{\gamma_k}\left(\int_0^1 \frac{1-2v}{\ell(\|\nabla f(x_k)\| + \|\nabla f(x_k)\|\,v)}dv\right)\|\nabla f(x_{k+1}) - \nabla f(x_k)\|^2$$

Since $\ell$ is increasing, we obtain

$$\int_0^1 \frac{1-2v}{\ell(\|\nabla f(x_k)\| + \|\nabla f(x_k)\|\,v)}dv \geq 0$$

and

$$\|\nabla f(x_{k+1})\|^2 \leq \|\nabla f(x_k)\|^2.$$

$\square$

## N. Proof of Corollary 8.4

*Proof.* For all $\delta > 0$, using Theorem 8.3, Corollary 5.2, and the fact that $\sup_{x \geq 0} \psi_2(x) > 0$, there exists $\bar{T}_1(\ell, \Delta, \delta)$ large enough, which depends only on $\ell$, $\Delta$, and $\delta$, such that $\|\nabla f(x_{\bar{T}_1})\| \leq \delta$. Since $\ell$ is continuous, there exists $\bar{\delta}(\ell) > 0$, which depends only on $\ell$, such that $\ell(2\bar{\delta}(\ell)) \leq 2\ell(0)$. Taking $\bar{T} \equiv \bar{T}(\ell, \Delta) := \bar{T}_1(\ell, \Delta, \bar{\delta}(\ell))$, we get $\|\nabla f(x_{\bar{T}})\| \leq \bar{\delta}(\ell)$ and

$$(15) \leq \bar{T} + \frac{\ell(2\bar{\delta}(\ell))R^2}{2\varepsilon} \leq \bar{T} + \frac{\ell(0)R^2}{\varepsilon}.$$

Thus Algorithm 1 converges after $\frac{\ell(0)R^2}{\varepsilon} + \bar{T}(\ell, \Delta)$ iterations because it converges after (15) iterations (Theorem 8.1).

Additionally, we know that

$$(15) \leq \bar{T}(\|\nabla f(x_0)\|) + \frac{\ell(2\|\nabla f(x_0)\|)R^2}{2\varepsilon} = \frac{\ell(2\|\nabla f(x_0)\|)R^2}{2\varepsilon},$$

where $\bar{T}(\|\nabla f(x_0)\|) = 0$ because it is required zero iterations to find a point such that the norm of a gradient is $\|\nabla f(x_0)\|$ (the starting point satisfies this criterion). In total, Algorithm 1 converges after at most

$$\min\left\{\frac{\ell(0)R^2}{\varepsilon} + \bar{T}(\ell, \Delta), \frac{\ell(2\|\nabla f(x_0)\|)R^2}{2\varepsilon}\right\} \leq \frac{\ell(0)R^2}{\varepsilon} + \min\left\{\bar{T}(\ell, \Delta), \frac{\ell(2\|\nabla f(x_0)\|)R^2}{2\varepsilon}\right\}$$

iterations. $\qquad\square$

## O. Proof of Corollary 8.5

*Proof.* Using Theorem 5.1 and Theorem 8.3, we get

$$\|\nabla f(x_{T-1})\|^2 \leq \frac{4\ell(2\|\nabla f(x_0)\|)\Delta}{T}.$$

Thus, for all $\delta > 0$, one can take $\bar{T}_1(\ell, \Delta, \|\nabla f(x_0)\|, \delta) = \frac{4\ell(2\|\nabla f(x_0)\|)\Delta}{\delta}$ to ensure that $\|\nabla f(x_{\bar{T}_1})\| \leq \delta$. Since $\ell$ is continuous, there exists $\bar{\delta}(\ell) > 0$, which depends only on $\ell$, such that $\ell(2\bar{\delta}(\ell)) \leq 2\ell(0)$. Taking $\bar{T} \equiv \bar{T}(\ell, \Delta, \|\nabla f(x_0)\|) := \bar{T}_1(\ell, \Delta, \|\nabla f(x_0)\|, \bar{\delta}(\ell))$, we get $\|\nabla f(x_{\bar{T}})\| \leq \bar{\delta}(\ell)$ and

$$(15) \leq \bar{T} + \frac{\ell(2\bar{\delta}(\ell))R^2}{2\varepsilon} \leq \bar{T} + \frac{\ell(0)R^2}{\varepsilon}.$$

Thus Algorithm 1 converges after (19) iterations because it converges after (15) iterations (Theorem 8.1). One can get the second term in the $\min$ using the same reasoning as at the end of the proof of Corollary 8.4. $\qquad\square$

## P. Proof of Theorem 9.2

Let us recall the standard result regarding the large derivation of the sum of i.i.d. random vectors.

**Lemma P.1** (Simplified Version of Theorem 2.1 from (Juditsky & Nemirovski, 2008)). *Assume that $\{\eta_i\}_{i=1}^m$ are i.i.d. random vectors such that $\eta_i \in \mathbb{R}^d$, $\mathbb{E}[\eta_i] = 0$, and $\mathbb{E}\left[\exp\left(\|\eta_i\|^2/\sigma^2\right)\right] \leq \exp(1)$ for all $i \in [n]$. Then*

$$\mathbb{P}\left(\left\|\sum_{i=1}^m \eta_i\right\| \geq \sqrt{2}(1+\lambda)\sqrt{m}\sigma\right) \leq \exp(-\lambda^2/3)$$

*for all $\lambda \geq 0$.*

We now proof the main theorem:

*Proof.* Using Lemma P.1, we get

$$\mathbb{P}\left(\left\|\frac{1}{B}\sum_{j=1}^B(\nabla f(x_k; \xi_{kj}) - \nabla f(x_k))\right\| \geq \sqrt{2}(1+\lambda)\sigma/\sqrt{B}\right)$$

$$= \mathbb{E}\left[\mathbb{P}\left(\left\|\frac{1}{B}\sum_{j=1}^B(\nabla f(x_k; \xi_{kj}) - \nabla f(x_k))\right\| \geq \sqrt{2}(1+\lambda)\sigma/\sqrt{B}\Big|x_k\right)\right] \leq \exp(-\lambda^2/3)$$

for all $k \geq 0$, where $\mathbb{P}(\cdot|x_k)$ is the probability conditioned on $x_k$. Using the union bound, we obtain

$$\mathbb{P}\left(\bigcup_{k=0}^{T-1}\left\{\left\|\frac{1}{B}\sum_{j=1}^B(\nabla f(x_k; \xi_{kj}) - \nabla f(x_k))\right\| \geq \sqrt{2}(1+\lambda)\sigma/\sqrt{B}\right\}\right)$$

$$\leq \sum_{k=0}^{T-1} \mathbb{P}\left(\left\|\frac{1}{B}\sum_{j=1}^{B}(\nabla f(x_k;\xi_{kj}) - \nabla f(x_k))\right\| \geq \sqrt{2}(1+\lambda)\sigma/\sqrt{B}\right) \leq T\exp(-\lambda^2/3).$$

Taking $\lambda = \sqrt{3\log(T/\delta)}$ and $B = \max\left\{\left\lceil\frac{32\left(1+\sqrt{3\log(T/\delta)}\right)^2\sigma^2}{\varepsilon}\right\rceil, 1\right\}$, we can conclude that with probability $1-\delta$,

$$\|g_k - \nabla f(x_k)\| = \left\|\frac{1}{B}\sum_{j=1}^{B}(\nabla f(x_k;\xi_{kj}) - \nabla f(x_k))\right\| \leq \frac{\sqrt{\varepsilon}}{4} \tag{35}$$

and

$$\|\nabla f(x_k)\| - \frac{\sqrt{\varepsilon}}{4} \leq \|g_k\| \leq \|\nabla f(x_k)\| + \frac{\sqrt{\varepsilon}}{4} \tag{36}$$

for all $k \in \{0,\ldots,T-1\}$, where we use the triangle inequality.

To simplify the notation, we assume that all subsequent derivations hold with probability $1-\delta$, omitting the explicit mention of it each time. Using proof by contradiction, we now prove that there exists $k \in \{0,\ldots,T-1\}$ such that $\|\nabla f(x_k)\|^2 \leq \varepsilon$. Assume that

$$\|\nabla f(x_k)\|^2 > \varepsilon \tag{37}$$

for all $k \in \{0,\ldots,T-1\}$. Then

$$\|g_k - \nabla f(x_k)\| \leq \frac{\|\nabla f(x_k)\|}{4} \tag{38}$$

and

$$\frac{3}{4}\|\nabla f(x_k)\| \leq \|g_k\| \leq \frac{5}{4}\|\nabla f(x_k)\| \tag{39}$$

from (35) and (36). Since $\ell$ is increasing, we obtain

$$\gamma_k\|g_k\| = \frac{1}{5r}\int_0^1 \frac{\|g_k\|\,dv}{\ell(\|g_k\| + \|g_k\|\,v)}$$
$$\overset{(39)}{\leq} \frac{1}{4}\int_0^1 \frac{\|\nabla f(x_k)\|\,dv}{r \times \ell\left(\frac{1}{2}\|\nabla f(x_k)\| + \frac{1}{2}\|\nabla f(x_k)\|\,v\right)}.$$

By the definition of the ratio $r$, we get $r \geq \frac{\ell(\|\nabla f(x_k)\| + \|\nabla f(x_k)\|v)}{\ell(\frac{1}{2}\|\nabla f(x_k)\| + \frac{1}{2}\|\nabla f(x_k)\|v)}$. Therefore

$$\gamma_k\|g_k\| \leq \frac{1}{4}\int_0^1 \frac{\|\nabla f(x_k)\|\,dv}{\ell(\|\nabla f(x_k)\| + \|\nabla f(x_k)\|\,v)}$$
$$\leq \int_0^1 \frac{\frac{1}{4}\|\nabla f(x_k)\|\,dv}{\ell\left(\|\nabla f(x_k)\| + \frac{1}{4}\|\nabla f(x_k)\|\,v\right)} = q(1/4\,\|\nabla f(x_k)\|\,;\|\nabla f(x_k)\|), \tag{40}$$

where $q$ is defined in Definition 4.1. Notice that $x_{k+1} = x_k - \gamma_k g_k = x_k - \gamma_k\|g_k\|\frac{g_k}{\|g_k\|}$ and $\gamma_k\|g_k\| \leq q(\frac{1}{4}\|\nabla f(x_k)\|\,;\|\nabla f(x_k)\|) < q_{\max}(\|\nabla f(x_k)\|)$. Thus, $x_{k+1} \in \mathcal{X}$ due to Lemma D.2 and we can use Lemma 4.5 to obtain

$$f(x_{k+1}) \leq f(x_k) - \gamma_k\langle\nabla f(x_k), g_k\rangle + \int_0^{\gamma_k\|g_k\|} q^{-1}(\tau;\|\nabla f(x_k)\|)d\tau$$

$$= f(x_k) - \gamma_k\|\nabla f(x_k)\|^2 - \gamma_k\langle\nabla f(x_k), g_k - \nabla f(x_k)\rangle + \int_0^{\gamma_k\|g_k\|} q^{-1}(\tau;\|\nabla f(x_k)\|)d\tau$$

$$\overset{\text{C-S}}{\leq} f(x_k) - \gamma_k \|\nabla f(x_k)\|^2 + \gamma_k \|\nabla f(x_k)\| \|g_k - \nabla f(x_k)\| + \int_0^{\gamma_k \|g_k\|} q^{-1}(\tau; \|\nabla f(x_k)\|) d\tau$$

$$\overset{(38)}{\leq} f(x_k) - \frac{3\gamma_k}{4} \|\nabla f(x_k)\|^2 + \int_0^{\gamma_k \|g_k\|} q^{-1}(\tau; \|\nabla f(x_k)\|) d\tau.$$

Since $q^{-1}$ is increasing (Propostion 4.2), we have

$$f(x_{k+1}) \leq f(x_k) - \frac{3\gamma_k}{4} \|\nabla f(x_k)\|^2 + \gamma_k \|g_k\| q^{-1}(\gamma_k \|g_k\|; \|\nabla f(x_k)\|)$$

$$\overset{(40)}{\leq} f(x_k) - \frac{3\gamma_k}{4} \|\nabla f(x_k)\|^2 + \frac{\gamma_k}{4} \|g_k\| \|\nabla f(x_k)\|$$

$$\overset{(39)}{\leq} f(x_k) - \frac{3\gamma_k}{4} \|\nabla f(x_k)\|^2 + \frac{5\gamma_k}{16} \|\nabla f(x_k)\|^2$$

$$\leq f(x_k) - \frac{\gamma_k}{4} \|\nabla f(x_k)\|^2.$$

It is left to sum this inequality for $k = \{0, \ldots, T-1\}$ and use Assumption 6.1 to get

$$\frac{1}{T} \sum_{k=0}^{T-1} \gamma_k \|\nabla f(x_k)\|^2 \leq \frac{4\Delta}{T}$$

and

$$\min_{k \in \{0, \ldots, T-1\}} \gamma_k \|\nabla f(x_k)\|^2 \leq \frac{4\Delta}{T}.$$

Due to Remark 4.7 and (39),

$$\gamma_k \geq \frac{1}{5r\ell(3\|\nabla f(x_k)\|)}$$

and

$$\min_{k \in \{0, \ldots, T-1\}} \frac{(\frac{3}{2}\|\nabla f(x_k)\|)^2}{\ell(3\|\nabla f(x_k)\|)} \leq r \times \frac{45\Delta}{T}. \tag{41}$$

Since (41) holds, we conclude that $\min_{k \in \{0, \ldots, T-1\}} \|\nabla f(x_k)\|^2 \leq \varepsilon$ due the choice of $T$ specified in the theorem. This result contradicts (37). $\qquad \square$

# Q. Function $-\mu x + e^{L_1 x}$ is $(L_1 \mu, L_1)$–smooth

In this section, we prove that the convex function $f : \mathbb{R} \to \mathbb{R}$ such that $f(x) = -\mu x + e^{L_1 x}$ is $(L_1 \mu, L_1)$–smooth. Indeed,

$$\|\nabla^2 f(x)\| = L_1^2 e^{L_1 x}.$$

Then

$$\|\nabla^2 f(x)\| = L_1^2 e^{L_1 x} \leq L_1 \mu + L_1 |L_1 e^{L_1 x} - \mu| \leq L_1 \mu + L_1 \|\nabla f(x)\|$$

because

$$\|\nabla f(x)\|^2 = (L_1 e^{L_1 x} - \mu)^2.$$

Finally,

$$\|\nabla^2 f(x)\| \leq L_1 \mu + L_1 \|\nabla f(x)\|.$$

At the same time, for any $A \geq 0$, there exists $x$ such that $L_1 e^{L_1 x} \geq 2\mu + \frac{2A}{L_1}$. We get $\frac{L_1}{2} e^{L_1 x} \leq \|\nabla f(x)\| \leq L_1 e^{L_1 x}$ and

$$A + 2L_1 \|\nabla f(x)\| \geq \|\nabla^2 f(x)\| = L_1^2 e^{L_1 x} \geq A + \frac{L_1}{2} \|\nabla f(x)\|.$$

Thus, the second constant in the assumption can be improved by at most of factor $4$.

