# OpenReview forum: "Toward a Unified Theory of Gradient Descent under Generalized Smoothness"
_ICML.cc/2025/Conference — ICML 2025 poster_

### Official Review · Reviewer_fNss · 2025-02-17

**Overall Recommendation:** 4

**Summary:**

---------
## Update after rebuttal

The authors showed that the issue in the proof can be fixed. I checked the corrected proof and it looks good to me, so I'm raising my score.

-------------


This paper studies the convergence of gradients descent under a generalized assumption on the objective $L$-smoothness. Assuming that the Hessian of the objective $f$ satisfies $\Vert \nabla^2 f(x)\Vert \le \ell (\Vert \nabla f(x)\Vert)$, where $\ell$ is a nondecreasing positive locally Lipchitz function, the authors show that with the stepsize $\gamma_k = $. The authors give a few good motivating toy examples such as classification with a linear two-layer network with a single parameter in each layer. Even this extremely simplified  example is not covered by previous theory, while the new work gives result if we choose $\ell(t) = L_0 + L_1t^\rho$ with some $\rho\ge 2$ (previous work only covers $\rho < 2$). And even in the case where the previous work does cover examples, the new stepsize rule leads to a superior  complexity.

The authors study nonconvex and convex problems. There are certain limitations to the results and one of the results is wrong (see the Theoretical Claims section of my review). The results for the case where $\ell$ is sub-quadratic have been already covered by the previous work, even though with a worse complexity, while the super-quadratic $\ell$ requires an assumption on bounded gradients. So while I found some of the small facts discovered in the paper to be elegant, the paper feels a bit inconclusive and doesn't seem to solve the problem of minimizing the functions advertised in the motivation.

The empirical results are very small and are only provided in Appendix A.

To conclude, the paper has very solid motivation, but the results are a bit underwhelming, especially taking into account one of them is wrong. I think removing the wrong part wouldn't be too difficult.

**Claims And Evidence:**

The claims are supported with evidence, I do not see any issue with that, and I'm pretty sure the authors willl be able to adjust them after removing the wrong part of the paper.

**Essential References Not Discussed:**

One key paper on the topic has been missed by the authors: Xie et al., "Gradient-Variation Online Learning under Generalized Smoothness". The paper considers online learning under a time-varying function $\ell_t$ in the generalized $\ell$-smoothness assumption.

Otherwise, the essential references seem to be already in the paper.

**Experimental Designs Or Analyses:**

The experiments are only performed on simplified 1-dimensional functions and reported in the appendix providing the number of iterations needed to reach a target accuracy. I think it's fair to say they don't constitute a solid contribution and the paper should be viewed as purely theoretical.

**Methods And Evaluation Criteria:**

There is no issues with methods and evulations.

**Other Comments Or Suggestions:**

When discussing related work and complexities, the proper definition of complexity is missing, which is especially important in the nonconvex case since some other papers discuss the complexity if getting $\Vert \nabla f(x) \Vert \le \varepsilon$ rather than $\Vert \nabla f(x) \Vert^2 \le \varepsilon$. Furthermore, the authors need to make it clear they discuss the best iterate rather than the last iterate or the average iterate.
I think the section structure is a bit sub-optimal. I'd suggest making sections 2, 3, and 4 subsections of section 1. Section 5, 6 could be unified as sections on general theory, while the rest except for the conclusion could be combined into a section on convergence.
Typo on page 4: "it is know that".
Typo on page 5: "it seems that this infeasible because to find an explicit formula of the optimal step size using (8)."
Typo on page 5: "one easily calculate $\gamma_k$".
Typo in equation (11): the gradient norm power is $p$ instead of $\rho$. Same typo is made in equation (28).
Typo on page 6: "If $\ell(s) = L_0 + L_1s^\rho$ for $p > 2$".
Typoe on page 6: "finds an $\varepsilon$–stationary after"
The way Assumption 9.1 is formulated is a bit weird as it seems to ask for non-uniqueness of $x_*$, which the authors don't need.

**Other Strengths And Weaknesses:**

I found it very easy to go through the paper and I'd commend the authors for the presentation of their results.

**Questions For Authors:**

1. The authors explain in Remark 6.7 that one can use the stepsize $\gamma_k = \frac{1}{\ell(2\Vert \nabla f(x_k)\Vert)}$ but it would lead to "a less tight final result". Can the authors clarifiy how much worse it's gonna be? From the practical perspective, the rule without integration seems to be a lot more appealing and relying on it would make the paper even stronger. Since (33) alreasy uses $\ell(2\Vert \nabla f(x_k)\Vert)$, it appears that the simpler choice would work at least for convex problems.

2. Can the authors provide an example of how Simpson's rule can be used on a specific example? Taking an example from Section 3 would be particularly illustrative.

**Relation To Broader Scientific Literature:**

The paper extends the prior work on generalized $L$-smoothness assumptions and provides a new stepsize that, to the best of my knowledge, hasn't been considered before. I think the contributions are novel and will be of interest to the optimization community.

**Theoretical Claims:**

The guarantees in this paper can be split it into several categories:
1. General **nonconvex** problems with $\ell(x)$ being **sub**-quadratic. This part looks **correct**, although not very interesting due to prior work already covering it.
2. General **nonconvex** problems with $\ell(x)$ potentially being **super**-quadratic. This one looks **correct**, but it requires an extra assumption that the gradient is bounded.
3. **Convex** problems with $\ell(x)$ being **sub**-quadratic. There is a **mistake** in the proof (see below).
4. **Convex** problems with $\ell(x)$ potentially being **super**-quadratic. This alternative approach seems to be correct as it uses a different proof technique.
5. **SGD** theory. I checked some steps in the proofs and they looked good to me, they follow the standard steps in high probability bounds. The convergence result requires large batch sizes, which is an issue present in a lot of previous papers on $(L_0, L_1)$-smoothness. It was shown in the work of Koloskova et al. that it's not surprising, clipping biases SGD and it doesn't converge in general. I see this as a small extension of the deterministic results.

## Mistake in the proof
In Appendix K, the authors use $x+y\le 2\max(x, y)$ in the wrong way. They conclude from $\min_k \frac{f_k}{x+y}\le \frac{R^2}{T+1}$ that  $\min_k \max(\frac{f_k}{2x}, \frac{f_k}{2y})\le \frac{R^2}{T+1}$, whereas the correct bound would be $\min_k \min(\frac{f_k}{2x}, \frac{f_k}{2y})\le \frac{R^2}{T+1}$, i.e., with $\min$ instead of $\max$ since $\frac{1}{\max(x, y)} = \min (\frac{1}{x}, \frac{1}{y})$. Unfortunately, I think it's a fundamental issue and can't be easily fixed.

---

> ### Author Rebuttal · Authors · 2025-03-27
>
> Thank you for the review!
>
> We now start with "Mistake in the proof." Thank you again for spotting a typo in the paper. **We will try to explain that this is a typo rather than a mistake because it is sufficient to fix "max" to "min," and nothing else should be changed.** Let us clarify this part. Both we and the reviewer agree that the right inequality is
>
> $$\min_{k \in \{0, \dots, T\}}  \min\left[\frac{f_k}{2 L_0}, \frac{1}{2 L_1 f_k^{\rho - 1}\left(\frac{2 \sqrt{T + 1}}{R}\right)^{\rho}}\right]
> \leq \frac{R^2}{T + 1}.$$ Let us fix $T$ as in the paper
> $$T = \max\left[\frac{2 L_0 R^2}{\varepsilon}, \frac{16 L_1^{2 / (2 - \rho)} R^2}{\varepsilon^{2 (1 - \rho) / (2 - \rho)}}\right]$$
> (smallest integer larger than this. Also, we should take 16 instead of 4).
>
> Notice that
>
> $$\min_{k \in \{0, \dots, T\}}  \min\left[\frac{f_k}{2 L_0}, \frac{1}{2 L_1 f_k^{\rho - 1}\left(\frac{2 \sqrt{T + 1}}{R}\right)^{\rho}}\right] = \min\left[\frac{f_T}{2 L_0}, \frac{1}{2 L_1 (f_T)^{\rho - 1}\left(\frac{2 \sqrt{T + 1}}{R}\right)^{\rho}}\right] \leq \frac{R^2}{T + 1} (**)$$
>
> because the terms in min are non-decreasing functions of $f_k$ for all $0 \leq \rho \leq 1$ and $f_T = \min_{ k \in [ 0, \dots, T ] } f_k$ (see Theorem 7.1). There are three options:
>
> 1) $\frac{f_T}{2 L_0} \leq \frac{1}{2 L_1 (f_T)^{\rho - 1}\left(\frac{2 \sqrt{T + 1}}{R}\right)^{\rho}}.$ Using (**), we get
>
> $$f_T \leq \frac{2 L_0 R^2}{T + 1} \leq \varepsilon,$$ where the last inequality due the choice of $T.$
>
> 2) $\frac{f_T}{2 L_0} > \frac{1}{2 L_1 (f_T)^{\rho - 1}\left(\frac{2 \sqrt{T + 1}}{R}\right)^{\rho}}$ and $\rho = 1.$ Using (**), we get
>
> $$(f_T)^{1 - \rho } \leq \frac{2^{1 + \rho} L_1 R^{2 - \rho}}{(T + 1)^{1 - \rho / 2}}$$
>
> and
>
> $$T + 1 \leq 16 L_1^2 R^2$$ because $\rho = 1,$ which can not be true due to the choice of $T.$ Thus, the second option is not possible after $T$ iterations.
>
> 3) $\frac{f_T}{2 L_0} > \frac{1}{2 L_1 (f_T)^{\rho - 1}\left(\frac{2 \sqrt{T + 1}}{R}\right)^{\rho}}$ and $\rho < 1.$ Using (**), we get
>
> $$(f_T)^{1 - \rho } \leq \frac{2^{1 + \rho} L_1 R^{2 - \rho}}{(T + 1)^{1 - \rho / 2}}.$$
>
> Since $T \geq \frac{16 L_1^{2 / (2 - \rho)} R^2}{\varepsilon^{2 (1 - \rho) / (2 - \rho)}},$ we get
>
> $$(f_T)^{1 - \rho } \leq \frac{2^{1 + \rho} L_1 R^{2 - \rho}}{(T + 1)^{1 - \rho / 2}} \leq \varepsilon^{1 - \rho}$$
>
> and
>
> $$f_T \leq \varepsilon$$
>
> because $\rho < 1.$
>
> In total, $f_T \leq \varepsilon$ for both possible options! We agree that these derivations are important, and we will add them to Section K. **The reviewer can see that there are no mistakes in Section K and only a small typo with "min" and "max".**
>
> Let us respond to other concerns:
>
> > One key paper on the topic has been missed by the authors: Xie et al., "Gradient-Variation Online Learning under Generalized Smoothness".
>
> Agree; we will add this paper to the discussion.
>
> > I'd suggest making sections 2, 3, and 4 subsections of section 1. Section 5, 6 could be unified as sections on general theory, while the rest except for the conclusion could be combined into a section on convergence.
>
> Agree; we will do it this way. See our discussion with Reviewer fNss.
>
> > Typos ...
>
> Thank you for spotting the typos!
>
> > The authors explain in Remark 6.7 that one can use the stepsize  but it would lead to "a less tight final result". Can the authors clarifiy how much worse it's gonna be?
>
> We believe that it can be significantly worse, and the new iteration complexity with $\ell(2 || \nabla f(x_k)||)$ can become $\sup_{s \geq 0} \ell(2 s) / \ell(s)$ bigger. For the $(\rho, L_0, L_1)$-smooth setup, the difference can be $2^{\rho}$ times. If $\rho$ is large, it is better to use our rule with the integral.
>
> > Can the authors provide an example of how Simpson's rule can be used on a specific example? Taking an example from Section 3 would be particularly illustrative.
>
> In our implementation, we use the standard `scipy` library as follows:
> ```python
> import scipy.integrate as integrate
>
> def find_step_size(ell, norm_grad):
>     return integrate.quad(lambda v: 1 / ell(norm_grad + norm_grad * v), 0, 1)[0]
> ```
>
> It is not necessary Simpson's rule, but some other numerical method from the Fortran library QUADPACK.
>
> > To conclude, the paper has very solid motivation, but the results are a bit underwhelming, especially taking into account one of them is wrong.
>
> We hope that our clarification of the typo improves the perception of our work. Note that our work provides new state-of-the-art theoretical complexities in both convex and nonconvex settings, which we believe constitutes an important contribution to the ICML community given the current interest in generalized smoothness.
>
> Thank you once again for spotting the typo!

---

> > ### Comment · Reviewer_fNss · 2025-04-02
> >
> > I thank the authors for showing how the issue can be fixed. I verified the details and it looks good to me. I'll increase my score for the paper accordingly.

---

### Official Review · Reviewer_4Yov · 2025-03-07

**Overall Recommendation:** 4

**Summary:**

This work improves the convergence rates of gradient descent on the $\ell$-generalized smooth problem for both nonconvex and convex settings by using a novel integral-based stepsize. Then it extends the results to stochastic gradient descent algorithm.

**Claims And Evidence:**

The improved convergence rates claimed by this work are well supported by theoretical results.

**Essential References Not Discussed:**

There are other works on $\ell$-smoothness, citing Li et al. (2024a). You could add them to the related works.

**Experimental Designs Or Analyses:**

The numerical examples and reproducibility (e.g. hyperparameter choices) look fine. It is strongly recommended to plot the learning curves $f(x_k)$.

**Methods And Evaluation Criteria:**

The numerical examples fit the generalized smooth condition well. The evaluation criterion I guess is the objective function $f(x_k)$, which is standard and reasonable. Please plot the learning curves which makes this criterion more clear.

**Other Comments Or Suggestions:**

(1) The suggestions about presentation above.

(2) The final paragraph of Section 2 "Related Works" could mention that $\ell$-smoothness generalizes $(L_0,L_1)$-smoothness. Also, there are other works on $\ell$-smoothness, citing Li et al. (2024a). You could add them.

(3) In Section 3, you could add machine learning application examples, for example, the examples in (Chen et. al. 2023). Also, are there any application examples that belong to $\ell$-smoothness but not $(L_0,L_1)$-smoothness (i.e., when $\ell(x)=\mathcal{O}(||x||^p)$ with $p\in(1,2)$ as $||x||\to+\infty$).

(4) The exact expression of Eq. (17) could be written in Table 2 for convenient comparison. If there is not enough space, you may consider denoting constants like $c=L_0^{\frac{\rho}{2+\rho}} \Delta^{\frac{\rho}{2+\rho}} L_1^{\frac{2}{2+\rho}} R^{\frac{4}{2+\rho}}$.

(5) The title of Section 5 could be "Assumptions for Nonconvex Setting". The title of Section 6 could be "Preliminary Theoretical Properties", since the main theoretical results are the convergence results in the consequent sections.

(6) You could explain how to compute such stepsize $\gamma_k$ as an integral form in practice.

(7) In Algorithm 2, you could explain that {$\xi_{kj}$}$_{j=1}^B$ are i.i.d. samples.

(8) In the equation right after (23), the middle step could use $g(v)$ instead of $g(t)$ to avoid confusing the two $t$.

(9) Lemma D.1 could define $U$ as $U(t,h)$ to be more clear.

(10) Right after Lemma P.1, "We now prove the main theorem".

**Other Strengths And Weaknesses:**

Strengths: The $\ell$-generalized smoothness studied in this problem is so far the most generalized smooth, which covers a lot of examples and applications. The theoretical results and proof techniques are correct and very novel. For example, this work uses inverse of the integral operator $q^{-1}$ which yields an integral-based bound on function decrease $f(x_{k+1})-f(x_k)$ and thus an integral-based stepsize. These further yield convergence results that improve the state of the arts.

Weakness mainly in presentation: (1) In ICML papers, usually Section 1 is introduction which introduces the studied problem, the drawback of existing works and the contribution of this work (move your Section 4 here) are missing in the Section 1. The final item of Section 4 could add "see Section xx". (2) The experimental results are usually shown by figure or table. In your case, it is strongly recommended to plot the learning curves $f(x_k)$. **I would like to raise my rating if you improve the presentation.**

**Questions For Authors:**

No questions now.

**Relation To Broader Scientific Literature:**

This work studies currently the most generalized smooth optimization problem ($\ell$-generalized smooth problem), with better convergence rates than the state of the arts in the other works on the same or less generalized smooth optimization problem.

**Theoretical Claims:**

By checking some key proofs, I believe the proofs are correct and novel.

---

> ### Author Rebuttal · Authors · 2025-03-27
>
> Thank you!
>
> > Please plot the learning curves which makes this criterion more clear.
>
> We've done this. We will add the following plots to the section with experiments: [figure](https://figicml.tiiny.site/)
>
> > Weakness mainly in presentation: (1) In ICML papers, usually Section 1 is introduction which introduces the studied problem, the drawback of existing works and the contribution of this work (move your Section 4 here) are missing in the Section 1.
>
> Thank you for the suggestions. We tried to follow the standard pattern: 1. Problem -> 2. Related Work -> 4. Contributions.  (yet, we added motivating examples to give more motivation to our contributions). **We will align our structure with the reviewer's expectations: i) we will swap "3. Motivating Examples" and "4. Contributions." Then, we will unify "1. Problem," "2. Related Work," and "3. Contributions." into one section called "1. Introduction." It can be easily done in the camera-ready version.**
>
> > The experimental results are usually shown by figure or table. In your case, it is strongly recommended to plot the learning curves
>
> **We will add the following plots to the section with experiments**: [figure](https://www.dropbox.com/scl/fi/xo2pr2cow8201ce4p75mj/fig.pdf?rlkey=cyhhm2wbvqrr2b21xqezzmjbi&e=1&st=7n46aqyx&dl=0)
>
> > The final paragraph of Section 2 "Related Works" could mention ... You could add them.
>
> > In Section 3, you could add machine learning application examples, for example, the examples in (Chen et. al. 2023). Also, are there any application examples t
>
> Agree. We will add these important papers to the overview.
>
> > The title of Section 5 could be "Assumptions for Nonconvex Setting". The title of Section 6 could be "Preliminary Theoretical Properties", since the main theoretical results are the convergence results in the consequent sections.
>
> Also agree; this title makes more sense.
>
> > You could explain how to compute such stepsize  as an integral form in practice.
>
> In our implementation, we use the standard `scipy` library as follows:
> ```python
> import scipy.integrate as integrate
>
> def find_step_size(ell, norm_grad):
>     return integrate.quad(lambda v: 1 / ell(norm_grad + norm_grad * v), 0, 1)[0]
> ```
>
> > In the equation right after (23), the middle step could use
>
> > Lemma D.1 could define
>
> > Right after Lemma P.1
>
> We will fix these problems.
>
> Thank you very much for the nice review, which helps to improve our work!

---

> > ### Comment · Reviewer_4Yov · 2025-04-04
> >
> > I have increased my rating to 4.
> >
> > For the plots: You could add captions to each plot, such as the corresponding objective function.

---

### Official Review · Reviewer_PL6G · 2025-03-09

**Overall Recommendation:** 4

**Summary:**

The paper discusses optimization under the generalized smoothness assumptions and show a choice of step size that improves the known convergence bounds in this setting. The paper also presents convergence rates in scenarios where previous work did not consider (e.g., $\rho\geq 2$).

# Update after rebuttal

In the rebuttal, the authors addressed my concerns about the computational cost and the accuracy needed in calculating the new step sizes. The is a good paper and I support in acceptance.

**Claims And Evidence:**

Yes, all of the theoretical claims in the paper are proved in the text or in the appendices.

**Essential References Not Discussed:**

No.

**Experimental Designs Or Analyses:**

No. I focused on the theoretical part of the paper.

**Methods And Evaluation Criteria:**

Yes, the paper is mostly theoretical.

**Other Comments Or Suggestions:**

N.A.

**Other Strengths And Weaknesses:**

Strengths:

1.The paper is well-written and clearly presented.
2. It establishes improved convergence rates for gradient descent under the generalized smoothness assumption in both convex and non-convex settings, which has been a topic of growing interest in recent years.
3. The approach of utilizing the function $q$ to reformulate the generalized smoothness conditions and derive a new step size is both novel and insightful.
4. The proposed step size allows the authors to obtain new theoretical results, also in the case of $\rho \geq 2$.

Weaknesses:

1.The fact that the new step size does not always have a closed-form expression could pose challenges for its practical implementation in real-world applications.
2. From a theoretical perspective, the additional computation required to determine the step size at each iteration may negatively impact the overall computational complexity compared to standard gradient descent.

**Questions For Authors:**

I have several questions regarding the computation of the new step size:

1. Is it necessary to compute the step size exactly in order to achieve the improved convergence rates? If not, how do the convergence bounds degrade when the step size is determined with an accuracy of $\delta$?

2. For a general $\ell$, what is the computational cost of computing the new step size with a precision of $\delta$?

**Relation To Broader Scientific Literature:**

The paper show convergence bounds that improve rates from previous work. Moreover, there are bounds for scenarios that was not studied in previous work. This improvements are achieved due to a different step size than considered in previous work.

**Theoretical Claims:**

No. I focused on reading the main text of the paper.

---

> ### Author Rebuttal · Authors · 2025-03-27
>
> Thank you!
>
> Let us respond to the weaknesses and questions:
>
> > The fact that the new step size does not always have a closed-form expression could pose challenges for its practical implementation in real-world applications.
>
> We now show how we implement it in the experiments and how it can be done in Python:
> ```python
> import scipy.integrate as integrate
>
> def find_step_size(ell, norm_grad):
>     return integrate.quad(lambda v: 1 / ell(norm_grad + norm_grad * v), 0, 1)[0]
> ```
> We believe that the implementation is straightforward and only requires the standard `scipy` library.
>
> > From a theoretical perspective, the additional computation required to determine the step size at each iteration may negatively impact the overall computational complexity compared to standard gradient descent.
>
> > For a general $\ell$, what is the computational cost of computing the new step size with a precision of $\delta$?
>
> We agree that the function above requires an additional call to `integrate.quad`. However, compared to the complexity of gradient computations, this operation is negligible. Calculating a numerical integral of a bounded and well-behaved function is a very simple one-dimensional numerical task.
>
> > Is it necessary to compute the step size exactly in order to achieve the improved convergence rates? If not, how do the convergence bounds degrade when the step size is determined with an accuracy $\delta$?
>
> This is a good question. We have not investigated it in detail, but the rationale is similar: computing the numerical integral in one dimension with very good accuracy is a very inexpensive operation. This operation is almost as simple as calculating $e^x$, division, and so on, which are also not computed with perfect precision. However, the errors arising from summation, division, $e^x,$ $\sin x,$ and other standard operations are typically ignored in practice due to the high level of accuracy.

---

### Official Review · Reviewer_eBsG · 2025-03-13

**Overall Recommendation:** 4

**Summary:**

This paper studies the performance of the Gradient Descent (GD) method under a generalized $\ell$-smoothness condition. The authors propose a universal step size applicable across different parameter choices. Using this step size, they improve existing theoretical results and establish new convergence guarantees for previously unexplored settings.

## update after rebuttal
Thanks the authors for answering my questions. I decide to keep my score.

**Claims And Evidence:**

Yes.

**Essential References Not Discussed:**

No.

**Experimental Designs Or Analyses:**

Yes. In fact, there are no experiments in the main content.

**Methods And Evaluation Criteria:**

Yes.

**Other Comments Or Suggestions:**

See above.

**Other Strengths And Weaknesses:**

Weaknesses:
* Can these results be extended to the constrained setting?
* It would strengthen the paper to include experiments demonstrating the universal applicability of the proposed step sizes across different problem settings.
* More details on the practical implementation of the step size computation would be beneficial.
* For experiments mentioned in Appendix A, it would be better to provide a figure to show the results.

Strengths:
* The paper provides a unified analysis and a universal step size for problems satisfying the $\ell$-smoothness condition, which encompasses many existing smoothness assumptions. Furthermore, the convergence results recover all known results as special cases.
* By leveraging the new step size, the authors improve the convergence rate of GD for certain function classes.
* The paper is well-written and easy to follow.

**Questions For Authors:**

See above.

**Relation To Broader Scientific Literature:**

See below.

**Theoretical Claims:**

Yes. I think it is correct.

---

> ### Author Rebuttal · Authors · 2025-03-27
>
> Thank you for the positive review!
>
> We would like to clarify the weaknesses:
>
> > Can these results be extended to the constrained setting?
>
> This is a good question. We have not yet explored this extension in depth; however, it appears that the constrained setting might be more challenging than in the $L$-smooth case. This is an important future work.
>
> > It would strengthen the paper to include experiments demonstrating the universal applicability of the proposed step sizes across different problem settings.
>
> Our main objective was to establish new theoretical results, supported by toy experiments. We acknowledge that there is significant potential for further research to explore the proposed approach in various practical and deep learning scenarios.
>
> > More details on the practical implementation of the step size computation would be beneficial.
>
>
> In our implementation, we use the standard `scipy` library as follows:
> ```python
> import scipy.integrate as integrate
>
> def find_step_size(ell, norm_grad):
>     return integrate.quad(lambda v: 1 / ell(norm_grad + norm_grad * v), 0, 1)[0]
> ```
>
> > For experiments mentioned in Appendix A, it would be better to provide a figure to show the results.
>
> We have prepared figures (see [figure](https://www.dropbox.com/scl/fi/xo2pr2cow8201ce4p75mj/fig.pdf?rlkey=cyhhm2wbvqrr2b21xqezzmjbi&e=1&st=7n46aqyx&dl=0)) that we will include in the camera-ready version of the paper.

---

### Decision · Program_Chairs · 2025-05-01

**Decision:**

Accept (poster)

**Comment:**

The paper derives the optimal step size for gradient descent under generalized smoothness assumptions. These assumptions have attracted much attention in the machine learning community. While the reviewers agreed that the work is well motivated and the results are new, the contribution is mainly theoretical, and the numerical advantage of the derived step size rule is far from obvious. The author(s) should take the reviewers' comments into account when revising the paper.